Review Article

EMBO
Molecular Medicine

# Human breast tissue engineering in health and disease

Maj-Britt Buchholz[1,2], Demi I Scheerman [ID][3], Riccardo Levato[3,4,5], Ellen J Wehrens[1,2,5] &
Anne C Rios [ID][1,2,5 ✉]

## Abstract

**The human mammary gland represents a highly organized and dynamic tissue, uniquely characterized by postnatal developmental cycles. During pregnancy and lactation, it undergoes extensive hormone-stimulated architectural remodeling, culminating in the formation of specialized structures for milk production to nourish offspring. Moreover, it carries significant health implications, due to the high prevalence of breast cancer. Therefore, gaining insight into the unique biology of the mammary gland can have implications for managing breast cancer and promoting the well-being of both women and infants. Tissue engineering techniques hold promise to narrow the translational gap between existing breast models and clinical outcomes. Here, we provide an overview of the current landscape of breast tissue engineering, outline key requirements, and the challenges to overcome for achieving more predictive human breast models. We propose methods to validate breast function and highlight preclinical applications for improved understanding and targeting of breast cancer. Beyond mammary gland physiology, representative human breast models can offer new insight into stem cell biology and developmental processes that could extend to other organs and clinical contexts.**

**Keywords** Mammary gland biology; Human breast modeling; Tissue engineering; Organ-on-a-chip; Bioprinting
**Subject Categories** Biotechnology & Synthetic Biology; Methods & Resources

## Introduction

The human breast performs the important function of nourishing offspring and promotes infant microbiome health and immune function. Despite these essential tasks, our knowledge of human mammary gland biology and pathology remains incomplete. This includes our understanding of breast cancer, one of the most frequent cancer-related deaths in women (Siegel et al, 2024). Representative in vitro models of breast tissue are needed to narrow this knowledge gap and support the development of new, effective therapies for breast cancer. Mouse models (Wagner, 2003) and human mammary epithelial cell (MEC) lines (Qu et al, 2015) have long been the gold standard for mammary gland research. Mouse models do provide a high degree of tissue complexity, and the possibility to study systemic interactions and treatment effects. However, they offer only limited representation of human breast function, due to interspecies differences in architecture, stromal composition, hormone levels, duration of lactation, and extent of involution (Medina, 2010). At the same time, the intricate connection of mammary gland function with its three-dimensional (3D) organization has long been recognized through pioneering work showing that alveoli formation and milk expression is tightly linked to the presence and distribution of extracellular matrix (ECM), biomechanical cues and architectural guidance (Barcellos-Hoff et al, 1989; Emerman and Pitelka, 1977; Petersen et al, 1992; Nelson et al, 2006), which cannot be recapitulated in human 2D cell cultures. This strongly emphasizes the need for more representative human in vitro models that recapitulate 3D architecture. Organoids have emerged as important in vitro 3D tools for human modeling, showing some level of self-organization (Brassard and Lutolf, 2019; Clevers, 2016; Drost and Clevers, 2018; Simian and Bissell, 2017; Sato et al, 2009). However, their higher-scale architectural complexity is still limited and variability in phenotype and cellular composition can hinder experimental, and especially drug-screening, applications (Rossi et al, 2018). Bioengineering offers a valuable technology for introducing more complex and reproducible structures and cellular organization in in vitro tissue models. Here, we review how different tissue engineering strategies can help to mimic the unique features of the human breast in vitro. We discuss the specific needs for ECM engineering and recapitulation of tissue-like architecture (Fig. 1) and highlight the potential of such engineered models for improving the clinical management of breast cancer. Finally, we address the maturation and validation strategies required to meet this goal (Fig. 1).

## Intricate interplay of cell composition, tissue architecture, and hormonal regulation

The mammary gland is composed of a ductal tree branching into secondary and tertiary ducts that end in terminal ductal lobular units (TDLUs), considered the functional units of the mature

[1]Princess Máxima Center for Pediatric Oncology, Utrecht, The Netherlands. [2]Oncode Institute, Utrecht, The Netherlands. [3]Department of Clinical Sciences, Faculty of Veterinary Medicine, Utrecht University, Utrecht, The Netherlands. [4]Department of Orthopedics, University Medical Center Utrecht, Utrecht University, Utrecht, The Netherlands. [5]These authors contributed equally: Riccardo Levato, Ellen J Wehrens, Anne C Rios. ✉E-mail: a.c.rios@prinsesmaximacentrum.nl

**Glossary**

**Biofabrication window**
describes a range of material properties that, to a certain degree, are both suitable for bioprinting and cell culture

**Bioprinting**
the process of depositing cells, biomaterials, and bioactive molecules in a spatially defined, computer-controlled, and layer-by-layer fashion, potentially into complex geometries

**Extrusion bioprinting**
bioprinting process during which a bioink is extruded from a syringe through a nozzle either by pneumatic or mechanical pressure, or screw-driven, and deposited onto a (potentially moving) surface

**Embedded extrusion-bioprinting**
extrusion-based bioprinting process that extrudes bioink into a support bath, providing structural support to facilitate the printing of hollow structures from soft bioinks

**Extracellular matrix**
network of extracellular proteins and bioactive molecules that surround the cells and provide structural and biochemical support, contributing to the mechanical properties of tissues

**Decellularized ECM**
cell-free ECM obtained through either physical, chemical, or enzymatical procedures to remove cells and lipids

**Micromolding**
the process of creating micro-topographies within 3D hydrogels is often executed using a (PDMS) stamp

**Organoid**
self-organizing 3D cell structure derived from single stem or progenitor cells that can be cultured and propagated long-term, while maintaining genetic stability and retaining phenotypical and functional features of the tissue of origin

**Organ-on-chip**
perfusable microphysiological system/microbioreactor that allows precise control over the conditions in which the contained cells are cultured, including the application of mechanical stimuli, such as flow

**Stereolitography-based bioprinting**
bioprinting process in which a liquid and photosensitive bioink is solidified layer-by-layer through selective spatial exposure to light of the appropriate wavelength

**Terminal endbud**
transient multilayered structure that directs the growth of the mammary gland duct during puberty

**Terminal ductal lobular unit**
fully matured functional unit of the mammary gland responsible for milk production

**Volumetric bioprinting**
bioprinting technique in which a vial filled with a photosensitive bioink is exposed to light projections from tomographic sections of the 3D model, while the vial is rotated

human mammary gland (Dontu and Ince, 2015). The ductal parts are characterized by an epithelial bilayer, with luminal cells composing the ductal inner layer and basal cells residing on the outer layer. Luminal cells are responsible for milk production, while basal cells can differentiate into myoepithelial cells that enable contraction and, thereby, discharge and transport of milk during lactation. In contrast to this bilayered organization in the duct, in the terminally differentiated TDLUs, luminal epithelial cells are multilayered to enable efficient milk production (Fig. 2) (Macias and Hinck, 2012; Slepicka et al, 2021). These structures are highly vascularized to ensure a sufficient supply of nutrients, hormones, and immune cells and embedded in a fibroblast-rich adipose stroma, which provides further architectural support and nutrient supply (Macias and Hinck, 2012; Dzięgelewska and Gajewska, 2018). Stromal fibroblasts critically contribute to mammary gland branching morphogenesis, and their diverse roles in this process and functional heterogeneity have been extensively reviewed elsewhere (Sumbal et al, 2021). While the mammary gland contains a population of tissue-resident, ductal macrophages involved in tissue integrity and homeostasis (Dawson et al, 2020), CSF-1-recruited macrophages also play a role and promote mammary gland branching morphogenesis (Nguyen and Pollard, 2002; Gouon-Evans et al, 2000). Adding further temporal complexity to this spatial and cellular regulation and organization, the functional maturation of the human breast is tightly regulated by stromal-produced growth factors, as well as hormonal cues provided by the pituitary gland and the ovaries (Slepicka et al, 2021; Hannan et al, 2023; Macias and Hinck, 2012). As an example of this complexity involving multiple factors and cell types, estrogen-stimulated macrophages enhance fibroblast proliferation and invasion through TNFα and matrix metalloproteinase (MMP) 9 secretion, which then provides a larger stromal bed to allow for epithelial cell proliferation and structure formation (Fleming et al, 2012). Thus,

the mammary gland is complexly regulated through an interplay of various cell types, their spatial organization, and hormonal and growth factor stimulation. Highly unique to the mammary gland, as opposed to other organs, this altogether cumulates in extensive tissue remodeling during postnatal developmental cycles (Fig. 2).

## Mammary gland developmental stages

A rudimentary ductal tree already establishes in utero guided by FGF and Wnt signaling (Slepicka et al, 2021). However, during puberty, aided by paracrine signaling of stromal cells and under the influence of multiple hormones, such as gonadotropin, estrogen, progesterone, growth hormone, and IGF1, elongation, branching, and bud formation take place (Slepicka et al, 2021; Macias and Hinck, 2012). This includes a role for eotaxin-recruited eosinophils (Gouon-Evans et al, 2000, 2002) and mast cells through serine proteases (Lilla and Werb, 2010) in this pubertal morphogenesis. During each estrous cycle, alveologenesis and mammary tissue expansion prepare the mammary gland for a potential pregnancy. In the case of pregnancy, progesterone, prolactin, IGF1, and insulin regulate MEC proliferation and terminal secretory differentiation of TDLUs into functional alveoli (Slepicka et al, 2021). Mediated by a postpartum drop in progesterone levels, the production of milk is initiated and oxytocin, as well as mechanical stimulation of a suckling newborn, enable milk secretion (Macias and Hinck, 2012). Upon cessation of breastfeeding, prolactin levels drop, milk-producing cells die and become phagocytosed by ductal macrophages (Dawson et al, 2020), and MMP-mediated tissue remodeling takes place (Hannan et al, 2023; Slepicka et al, 2021). Finally, an increase in leptin leads to adipogenesis, replacing the secretory epithelium with adipocytes (Zwick et al, 2018), completing this involution process and returning the mammary gland to its pre-pregnancy state (Hannan et al, 2023; Slepicka et al, 2021).

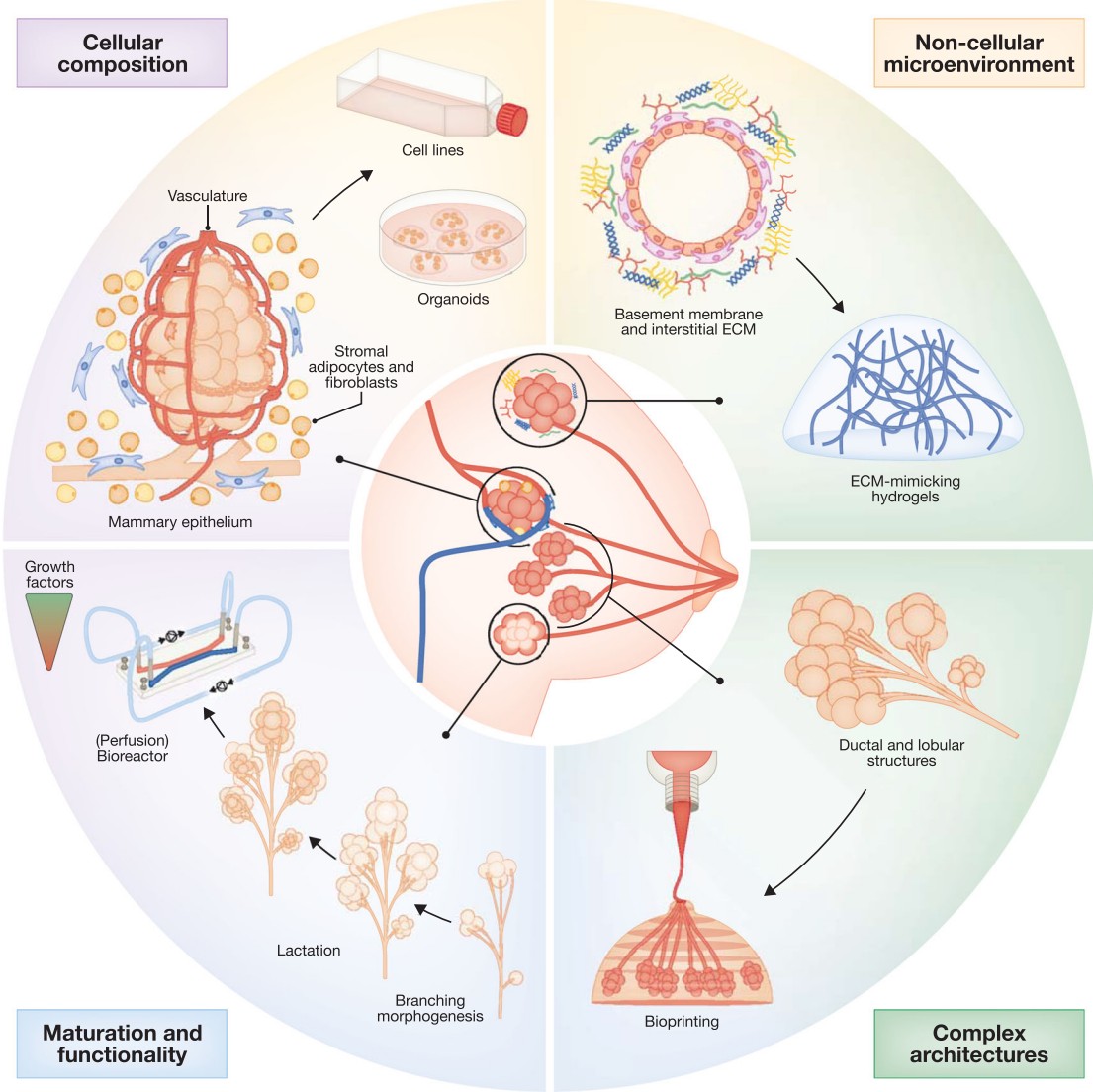

**Figure 1. Key features of mammary gland biology and how they could be recapitulated through tissue engineering technologies.**

The cellular compartments which need to be considered consist of the functional mammary epithelium comprised of luminal and basal epithelial cells, the stroma comprising mammary fibroblasts, adipocytes and immune cells, as well as the vasculature. These cellular compartments can be addressed using cell lines or primary cell sources such as patient/donor-derived organoids. The non-cellular microenvironment consists of a network of extracellular matrix components, which can be mimicked using specifically tailored hydrogels. Bioengineering techniques, such as bioprinting and organ-on-a-chip technology, enable the fabrication of recapitulative tissue architectures and help mimic crucial stimuli, such as flow and nutrient/hormone supply, to achieve full tissue maturation and functionality.

## Mammary gland extracellular matrix represents a complex and dynamic environment

Next to the cellular and structural changes described in the previous section and summarized in Fig. 2, the non-cellular compartment of the mammary gland strongly impacts its morphogenesis and functionality (Rauner et al, 2021) and is dynamically regulated as well (Schedin and Keely, 2011). It, thereby, forms a critical aspect to take into account in mammary gland modeling approaches to guide cell organization.

## Of mice and women

While limited compared to studies in mouse model systems, analysis on human breast ECM has been done for, example, by Gusterson et al looking at the positional relationship between myoepithelial cells and basement membrane (Gusterson et al, 1982) and by Keller et al comparing the spatial distribution of collagen-I in human breast tissue, human-derived decellularized breast tissue, collagen-1 hydrogels, and Matrigel. They could identify mesh-like and long fibrous ECM organization and loss of this specific texture in ductal carcinoma samples. This interconnected fibrous network

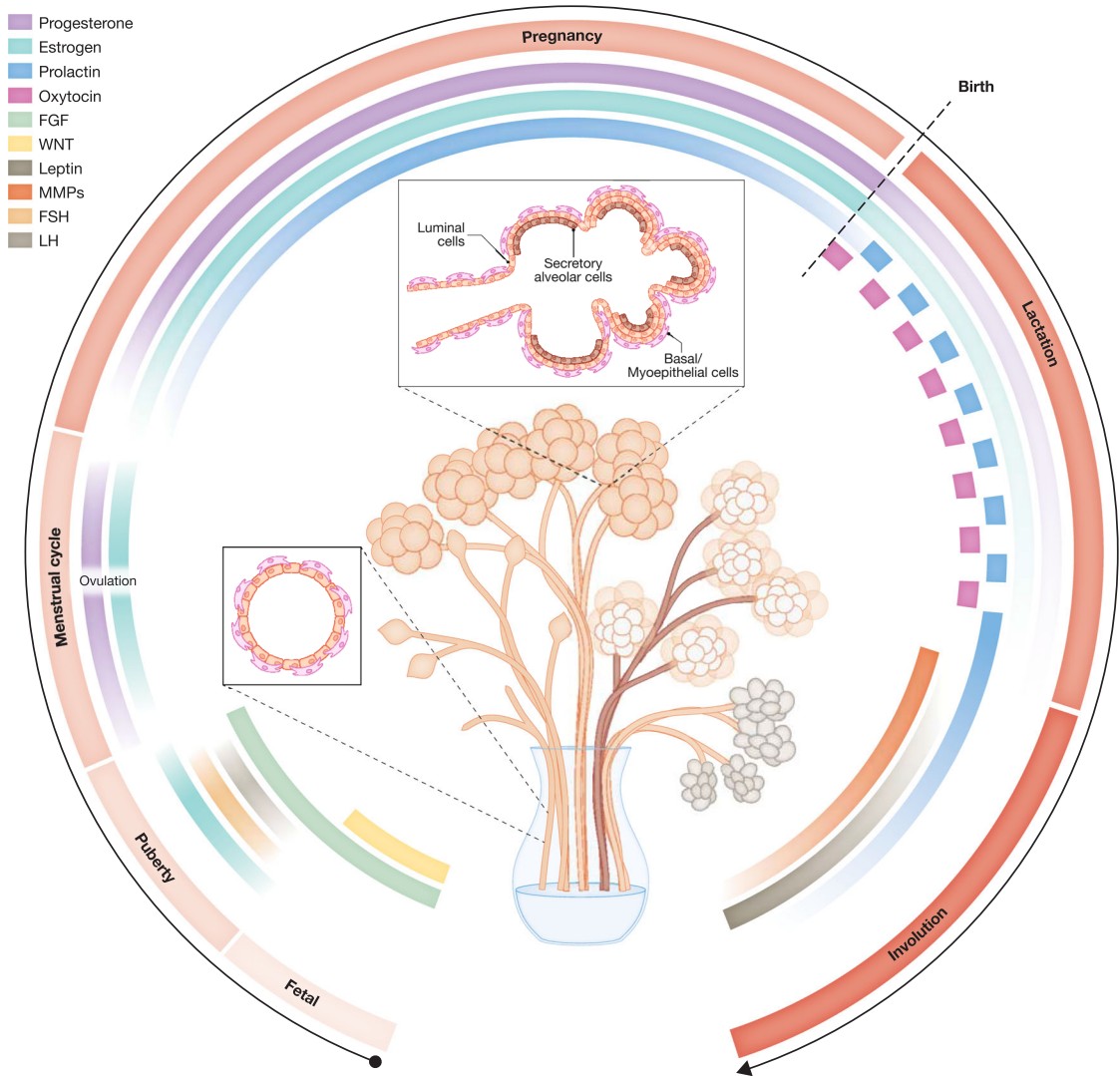

**Figure 2. Architectural structure of the mammary gland and how it is hormonally regulated during various cycles of development.**

A rudimental ductal tree comprised of a bilayered epithelium that already exists during fetal development, under the influence of various hormones, continues to expand during puberty, and undergoes dynamic changes throughout each menstrual cycle. In the case of pregnancy, rising levels of estrogen, progesterone, and prolactin lead to the maturation of the mammary gland into fully functional multilayered alveoli. After birth, sudden drops in estrogen and progesterone levels, as well as alternating stimulation through oxytocin and prolactin, enable milk production and secretion. Under the influence of MMPs and leptin, the mammary gland undergoes involution to a pre-pregnancy state after weaning.

was maintained in dECM-derived hydrogels, while interconnectivity could not be observed in regular collagen-1 hydrogels. In addition, they observed a 5.7-fold increase in ECM stiffness in tumor samples compared to normal breast tissue (Keller et al, 2021), a mechanical property of breast tissue that is commonly used as a parameter for cancer detection (Ramião et al, 2016). Also, in a healthy situation, the mechanical properties of breast tissue can show dynamic alterations. The collective motion of epithelial cells, for instance, induces local tension and mechanical response of the surrounding collagen, which plays a role in ductal elongation (Buchmann et al, 2021). However, the majority of our knowledge on breast ECM composition and function is derived from mouse data, as further detailed below. It is important to note that, despite many similarities, there are also substantial interspecies differences

in ECM composition, exemplified by the complete absence of intralobular stroma in mice, as opposed to humans (Dontu and Ince, 2015). This signifies a need to carefully consider the origin of ECM components for mammary gland modeling and assess whether data on ECM composition or spatial organization can be translated from mouse to human.

## Mammary ECM composition

The mammary epithelium is lined by a thin basement membrane mainly made up of collagen-IV and laminins (Yurchenco and Patton, 2009). This basement membrane, produced by myoepithelial cells (Gudjonsson et al, 2002) and stromal fibroblasts (Muschler and Streuli, 2010), is a key regulator of the integrin-mediated

polarized mammary epithelial phenotype. Particularly, laminin-III and collagen-IV are responsible for maintaining tissue polarity (Muschler and Streuli, 2010; Nelson and Bissell, 2006; Insua-Rodríguez and Oskarsson, 2016; Schedin and Keely, 2011). The surrounding ECM is characterized by a looser network of an interstitial fibrous collagen-rich matrix, including collagens-I and -III, proteoglycans, hyaluronic acid (HA), fibronectin, tenascins, and various matrix remodeling enzymes (e.g., MMPs) (Insua-Rodríguez and Oskarsson, 2016; Muschler and Streuli, 2010).

## Dynamic ECM changes during mammary gland development

The composition of the mammary ECM varies strongly during the different mammary gland developmental stages described above, as extensive remodeling takes place to allow the cells to branch from primary to secondary ducts, or to form TDLUs, and regress again during involution (Insua-Rodríguez and Oskarsson, 2016; Muschler and Streuli, 2010). For instance, when ductal elongation occurs during puberty, the basement membrane of terminal end buds (TEBs; transient structures found in the pubertal gland) is thinning, rich in HA, and does not contain collagen-I bundles (Schedin and Keely, 2011). The surrounding ECM presents with collagen fibers radiating away from these TEBs, facilitating tissue growth from the tips (Ingman et al, 2006). Finally, the basement membrane adjacent to ductal parts is actually rich in collagen-I, where no outgrowth occurs (Silberstein and Daniel, 1982). These findings in mice clearly demonstrate; (1) the various changes occurring in ECM composition during puberty, (2) that these alterations are spatially restricted, and (3) how this topography of dynamic ECM composition directs ductal elongation and branching. During lactation, generally less fibrous ECM proteins (e.g. collagens) can be observed, providing a compliant environment for MECs to fully mature, while during involution, these components are again increased (Schedin et al, 2007).

## Cancer-associated dynamic ECM alterations

ECM remodeling also takes place during breast cancer progression. Changes in protein make-up, such as through downregulation of laminin-III and upregulation of MMPs, lead to basement membrane disruption and degradation, causing a loss in epithelial polarity and enabling invasiveness of the tumor (Schedin and Keely, 2011; Nelson and Bissell, 2006; Muschler and Streuli, 2010; Insua-Rodríguez and Oskarsson, 2016). At the same time, laminin-332, which promotes tumor progression through anchorage-independent survival, is upregulated (Insua-Rodríguez and Oskarsson, 2016). Overall, breast tumorigenesis is accompanied by desmoplasia; increased matrix protein deposition (especially fibrillar collagens) and remodeling, resulting in an increase of ECM density and stiffness promoting metastatic behavior and correlating with poor treatment outcomes (Insua-Rodríguez and Oskarsson, 2016; Schedin and Keely, 2011). These changes in matrix composition and density during breast cancer progression, such as proteolysis of basement membrane proteins following MMP upregulation, share a lot of similarities to the changes occurring during involution, explaining the temporarily increased risk and poor prognosis of breast cancer postpartum (Lyons et al, 2011; McDaniel et al, 2006; Schedin et al, 2007). In contrast, during

lactation, deposition of fibrillar collagen-I is reduced, correlating with a reduced risk of breast cancer through breastfeeding, which decreases even more upon prolonged breastfeeding (4.3% for every 12 months of breastfeeding) (Maller et al, 2013; Takabatake et al, 2016; Stordal, 2023). Because of its many facets influencing mammary gland development and pathology, it is clear that exploiting the ECM to guide functional cell organization in in vitro models is not only reliant on the spatial availability of different matrix proteins, but also on their biomechanical properties and their ability to be dynamically remodeled.

# Guiding cellular self-organization in vitro through biomaterial modulation

## Multimaterial ECM

Various strategies have been implemented to culture normal and cancerous MECs in different natural ECM components, as briefly summarized in Table 1. While these approaches using different single ECM components were successful to some degree, it is becoming more and more evident, that relying on a single-material approach might not yield fully functional breast models. Therefore, the focus is shifting towards multimaterial approaches. An example is the work of Nguyen-Ngoc et al, who cultured murine primary MECs in Matrigel or a mix of Matrigel and collagen-I. (Nguyen-Ngoc et al, 2015). In mixed gels, they observed a more faithful recapitulation of branching and TEB formation (e.g., bifurcation and myoepithelial coverage during bud initiation/elongation) than in Matrigel, whereas collagen-I only induced an invasive phenotype that could be restored by the cells own basement membrane production. Extended branching and endbud formation could be reproduced for mouse mammary gland organoids (MGOs) in floating collagen-Matrigel gels (Qu et al, 2017), a culture method already pioneered by Emerman and Pitelka in 1977 (Emerman and Pitelka, 1977). However, this induced only little branching and bud elongation in human iPSC-derived MGOs, suggesting a requirement for a more complex ECM composition for human MECs. This was confirmed by Sokol et al, who cultured primary human MECs in a multimaterial matrix containing natural collagen-I, laminin, fibronectin, and hyaluronan, and observed self-organization, expansion, and differentiation into near-mature mammary tissues comprised of ductal and lobular morphologies with correct polarization (Sokol et al, 2016). Furthermore, the resulting structures responded to hormonal stimulation with lumen formation and induction of lipid droplet production. While even in this mixed gel, complex ductal and lobular morphologies did not occur very frequently, they were never observed in case Matrigel only, or only polymerized collagen was used to grow the cells. Rauner et al recently used this multimaterial matrix to yield human breast 3D structures through directed organogenesis rather than spontaneous self-assembly usually observed in BME (Rauner et al, 2023). Together, these results emphasize the impact of ECM complexity on cellular organization and function and indicate that the optimal ECM composition can differ between mouse and human breast tissue. Further knowledge on human ECM composition and its dynamic changes during the development of complex architectures will be necessary to mimic breast morphogenesis in vitro. Such a comprehensive map of the human breast ECM during different

**Table 1. Properties and functional outcomes of different ECM components on MECs.**

| Biomaterial | Properties | Cell type | Modality | Functional outcome | Reference |
|---|---|---|---|---|---|
| Matrigel/EHS | - Laminin-rich, contains collagen-IV and multiple other proteins (undefined composition)<br>- Highly bioactive<br>- Temperature-dependent polymerization<br>- Poor shape stability over time in culture<br>- High batch-to-batch variability | Mouse MECs | On top | Promotes mammary gene expression | (Li et al, 1987) |
| | | | | Alveoli and lumenized duct formation and milk production on floating, but not attached gels | (Barcellos-Hoff et al, 1989) |
| | | Human primary MECs Biopsy-derived human BC cells | Encapsulated and on top | Healthy and cancerous cells can be distinguished by their growth characteristics | (Petersen et al, 1992) |
| | | Human BC organoids | Encapsulated | BC organoids maintain morphology, hormone receptor status and treatment response of tumor of origin | (Sachs et al, 2018) |
| | | Human normal and cancerous organoids | | - Epithelial lineages are maintained in culture<br>- ER+ status can be maintained<br>- Protein expression patterns are maintained | (Rosenbluth et al, 2020) |
| Collagen-I | - Most abundant ECM protein<br>- pH-dependent polymerization<br>- Bioactive<br>- Poor shape stability over time in culture | Primary human breast and BC cells | Encapsulated | - Inducing protrusive migration in BC cells<br>- Inducing transient protrusive migration in healthy MECs until a basement membrane is produced | (Nguyen-Ngoc et al, 2012) |
| | | Mouse MECs | On top of floating gels | Milk protein expression only on top of floating gels | (Emerman and Pitelka, 1977) |
| | | Human MCF10A HMT3522 S-1 | | - MECs cultured in lower concentration 3D collagen gels organize into polarized acinar and ductal structures<br>- MECs in high concentration collagen gels lose polarization and instead become locally invasive | (Paszek et al, 2005) |
| | | Human breast organoids | Encapsulated | Plastic mechanical responses of collagen direct ductal elongations | (Buchmann et al, 2021) |
| Fibrous collagen-I | | Human MCF7 T-47D | Encapsulated | Promotes ER-independent tumor cell proliferation | (Reyes-Ramos et al, 2021) |
| Hyaluronic acid (HA) | - Bioactive<br>- Poor shape stability | Human MCF7 | | - Displays similar mechanical properties to breast tumor and can be degraded by cells<br>- Supports cell viability and proliferation<br>- Promotes greater migration/invasion abilities and tumorigenicity than 2D | (Suo et al, 2019) |
| Fibronectin | - Abundant in breast mesenchyme<br>- Can only form stable hydrogels when modified (e.g., PEGylation) | Human MCF7 | On top | Induces EMT, migration and invasion | (Li et al, 2017) |
| Alginate | - Bioinert<br>- Ionic polymerization<br>- Stable 3D structures over prolonged culture | Mouse mammary organoids | Encapsulated; microbeads | - Similar phenotypes and cell lineages in alginate bead compared to original tumor<br>- Luminal pressure influences drug response | (Fang et al, 2021) |
| Alginate-Matrigel | - Structurally instable<br>- Provides biochemical cues to the cells | Human MDA-MB231 | Encapsulated | Invasive phenotype in 50% Matrigel 50% alginate | (Cavo et al, 2018) |

developmental stages could guide biomaterial engineering to build near-physiological matrices for advanced in vitro modeling.

## Decellularized extracellular matrix

While multimaterial approaches show superiority for modeling human mammary gland morphogenesis, these approaches yield a random distribution of only a selection of ECM proteins. A possibility to retain not only tissue-specific native ECM composition, but also spatial arrangement is the use of tissue-derived decellularized ECM (dECM). Depending on tissue type and size, as well as desired application, tissue pieces are objected to a series of physical (e.g., freezing or hydrostatic pressure), chemical (e.g., Triton X-100 or SDS), and enzymatical (e.g., trypsin) procedures to remove cells and lipids, to obtain a cell-free ECM (Mendibil et al, 2020). Some of the more gentle approaches allow to retain the original tissue architecture, as demonstrated by Jin et al, who used human normal and cancerous tissue to produce decellularized scaffolds and reseed them with BC cells. While cancer-derived dECM induced epithelial- mesenchymal transition (EMT) and cancer cell proliferation, normal-derived dECM did not (Jin et al, 2019). This demonstrates a strong functional influence, which was also shown for age-dependent changes in dECM. Compared to young murine dECM, dECM derived from old mice was sufficient to induce cancer-like behavior in normal MECs and promote invasiveness in cancerous cells, demonstrating a role for the ECM in increased breast cancer risk with advanced age (Bahcecioglu et al, 2021). These profound functional consequences strongly argue that dECM provides the most accurate native tissue environment. However, because of interspecies differences, human tissue is needed to faithfully recapitulate the human breast ECM. While protocols are in place for processing human reduction mammoplasty and mastectomy material (LaBarge et al, 2013), not all labs will have the required ethical approvals in place and easy access to these tissue sources. In addition, although some decellularization methods maintain tissue architecture, they do not allow control over it, making these approaches less suitable for engineering purposes. Therefore, most tissue engineering technologies opt for more disruptive protocols to produce hydrogels from decellularized matrix proteins. Here, the composition and proportions of ECM components are maintained, but the structure and spatial distribution is lost (Mendibil et al, 2020). This could potentially be recovered by the utilization of biofabrication techniques. Additive manufacturing techniques, for instance, 3D bioprinting could reintroduce architecture through the spatially controlled deposition of hydrogel, but also living cells and molecules (Groll et al, 2016). However, when applying ECM components in such a bioprinting setting, their requirements for promoting optimal cell function often compete with the requirements posed by the fabrication modalities and desired shape fidelity. Many ECM components, such as collagen-I or laminin, are too soft to retain shape fidelity post printing at concentrations that allow for proper cell function, while many synthetic materials, such as poly(ethylene glycol) (PEG), which can easily be tailored to the desired mechanical properties, do not offer bioactive cues to promote cell viability and function. Therefore, many biofabrication approaches have been restricted to what has been termed the *Biofabrication Window* and moving out of this traditional window has been identified as one of the critical steps for the biofabrication field to

progress toward more representative human tissue models. For now, biofabrication strategies are stuck in a range of material properties (concentration, stiffness, crosslinking density) that allow for only moderate shape fidelity to also ensure cell viability, making concessions on both critical aspects for effective tissue modeling (Malda et al, 2013) (Fig. 3). Thus, additional developments are needed to yield printed tissue constructs that promote proper cell function, while maintaining high shape fidelity (Malda et al, 2013). Multiple approaches to achieve this have been suggested, for instance, combining bioinks with poor structural integrity, but high bioactivity with a mechanically supportive material, or support baths, to enhance their shape fidelity during printing (Malda et al, 2013; Levato et al, 2020). In addition, bioink design is applied to make synthetic highly printable materials more compatible with cell viability and function, for instance, through the addition of cell signaling sequences, or enzymatically cleavable sequences. Reversibly, the printability and shape fidelity of natural polymers that already possess bioactive moieties can be improved by introducing new crosslinking chemistries or rheological tuning (Levato et al, 2020; Malda et al, 2013).

## Development of bioprintable biomaterials

Zhang et al proposed a biocompatible gel made of oligo(ethylene glycol)-grafted polyisocyanides (PICs) as a synthetic hydrogel for murine MGO culture (Zhang et al, 2022, 2020). To improve bioactivity, the gel was functionalized with the cell adhesion peptide RGD. Organoid formation efficiency was shown to be dependent on stiffness; with softer gels promoting higher organoid formation, compared to stiffer gels. In contrast, the cellular composition was influenced by RGD-density and, thereby, the frequency of cell–matrix interactions; high-density RGD-gels led to an increased proportion of basal cells in MGO cultures (Zhang et al, 2020). This modularity of PIC-based materials, enabling optimization to fabrication needs, as well as biological requirements, makes them attractive materials for biofabrication (Sánchez et al, 2020). However, natural ECM components contain far more than just adhesive peptides to guide cell behavior. Identifying the full spectrum of tissue-specific bioactive sequences and their required spatial distributions to generate fully ECM-mimicking synthetic materials poses a challenging task. Therefore, the modification of natural polymers that already possess these properties is another path toward fabrication-suitable bioactive materials. This is, for instance, confirmed by Bock et al, who compared synthetic to natural—but modified—biomaterials for the growth of BC organoids (Bock et al, 2023). This work showed that the use of GelSH, thiolated gelatin, a natural collagen-derived polymer that can be crosslinked to enhance shape fidelity, resulted in the most complex and large growing BC organoids, as compared to traditionally used biomaterials. Moreover, the authors highlight future perspectives on the use of GelSH for precision medicine applications, where the gel stiffness can potentially be modulated to fit the tumor of origin (Bock et al, 2023). Protease-degradable crosslinkers have been introduced into an HA-based hydrogel to consider the need for cells to be able to remodel their microenvironment, which resulted in endothelial cells sprouting towards an angiogenic gradient (Song et al, 2018). Together, these approaches show the potential of modifying synthetic and natural polymers to improve tissue function during fabrication. However,

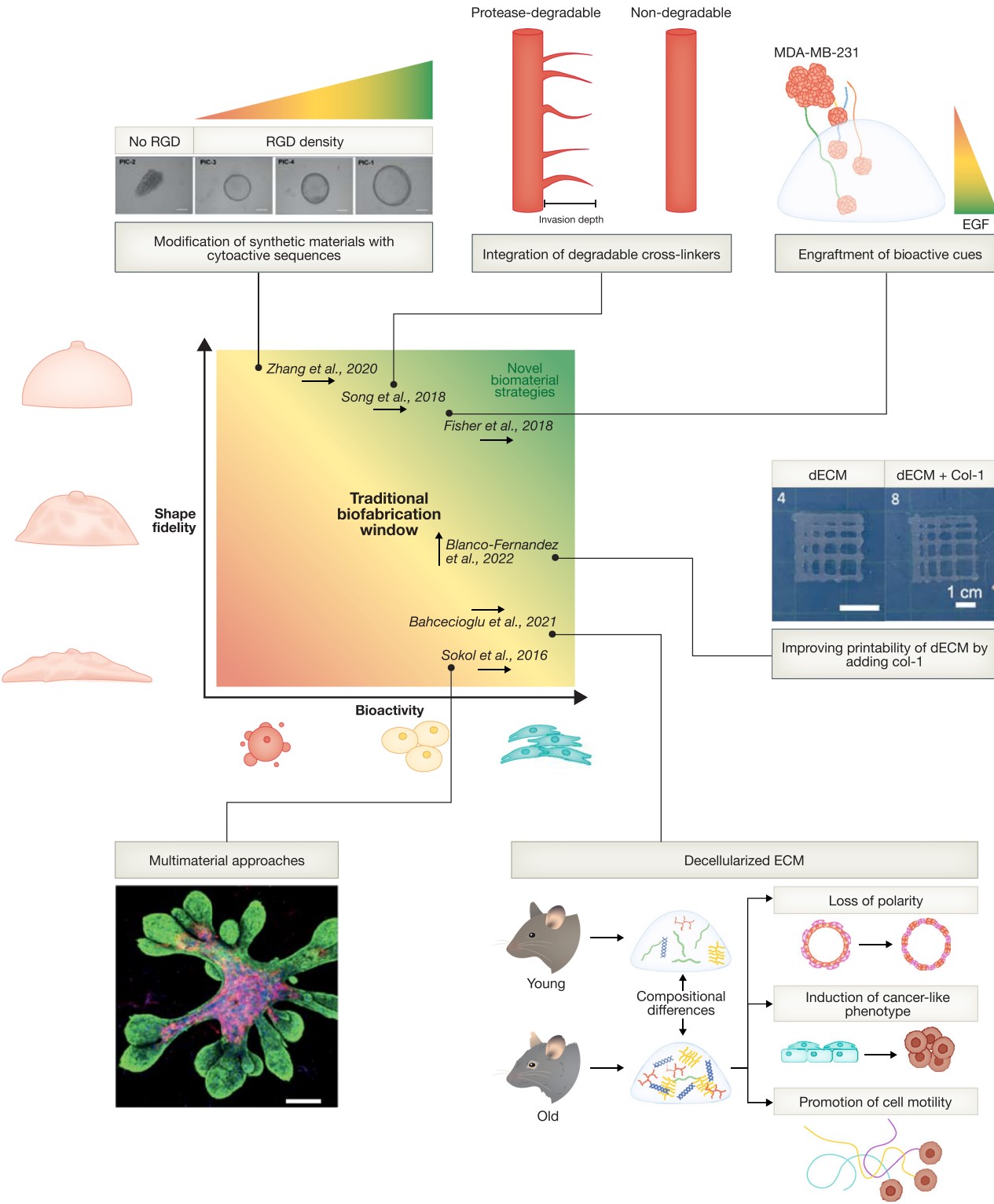

**Figure 3. Approaches to guide MEC behavior through biomaterial engineering.**

Shape-fidelity post-fabrication and cell function often impose opposing requirements on biomaterials. Strategies to achieve both, include the modification of materials with high shape fidelity to enhance bioactivity, or the modification of bioactive materials to enhance shape fidelity. Positioning within the Biofabrication Window does not imply qualitative classification of the used materials, but points towards the direction of improvement the presented research aims to achieve. Reproduced with permission from (Zhang et al, 2020) (Creative Commons Attribution License) and (Blanco-Fernandez et al, 2022; Sokol et al, 2016) (Creative Commons Attribution 4.0 International License; CC BY 4.0).

most of these approaches still make concessions between either printing accuracy or bioactivity, which has sparked the development of even more advanced biomaterials. For instance, Skardal et al composed bioinks compatible with extrusion-based bioprinting by combining dECM from different tissues with HA, gelatine-based hydrogels, and PEG-based crosslinkers, yielding tissue products of a wide range of stiffnesses that contained the bioactive components of the tissues' native ECM (Skardal et al, 2015). For breast tissue, a similar approach has been applied to improve the print fidelity of dECM by the addition of collagen-I (Blanco-Fernandez et al, 2022). Bioactive properties can also be advanced by the controlled release of, or grafting, of cell favorable components, such as growth factors. Selectively photo-grafting of vascular endothelial growth factor (VEGF) into a print made of gelatine-norbonene was able to direct sprouting of endothelial cells into the surrounding tissue (Falandt et al, 2023). Similarly, Fisher et al photografted EGF onto a HA-based hydrogel and showed that it preferentially directed BC cell invasion into the tissue in relation to the expression of EGF receptor (EGFR) on the cell lines used (Fisher et al, 2018). More recently, the generation of supramolecular materials that employ non-covalent bonds has gained much attention, due to the possibility of cells to break bonds to migrate and expand in the material, while shape fidelity can be maintained (Mihajlovic et al, 2021).

# Capturing native architecture with tissue engineering technology

## Geometrical constraints guide tissue development and function

Growing scientific evidence from various organs demonstrates that geometrical constraints, as well as the spatial distribution of cell-ECM and cell–cell interactions can guide (stem-) cell proliferation, migration, organization, and functional differentiation in vitro (Fig. 4) (Baker and Chen, 2012; Brandenberg and Lutolf, 2016; Brock et al, 2003; Callens et al, 2023; Levato et al, 2020). A prominent example is the use of topographically structured hydrogels to mimic the intestinal stem cell niche, leading to the functional differentiation of intestinal organoids without additional cues (Gjorevski et al, 2022). This demonstrates that shape alone can stir cells towards functional maturation and that such structure–function relationships can be modeled and exploited in vitro. Micromolding approaches have been applied to breast culture models to introduce shape constraints, using a stamp to produce microwells from soft gelatin (Cerchiari et al, 2015) (Fig. 5). For instance, Nelson et al controlled branching morphogenesis through micropatterning and found that the geometry of tubules dictates the position of branches, with branches occurring preferably at convex positions of the patterns, coinciding with the local minimum of autocrine inhibitory morphogens (Nelson et al, 2006) (Fig. 4). Using the same approach to control the location of branching, Lee et al found that the transcription factors Snail1, Snail2, and E47 are specifically upregulated at branching sites, helping to elucidate their role in promoting branching morphogenesis (Lee et al, 2011). It is, thus, clear that tissue architecture, is not only a result, but also a determinant of function, and mimicking this will be essential for the proper function of an in vitro model.

Experimental control over complex architectures will further aid in reproducibility, accessibility, and adaptability to specific research questions at hand. The next sections, therefore, focus on tissue engineering approaches to achieve more complex architectures and spatial control over cell and ECM components to guide cell organization and function.

## Breast-on-chip systems

Organ-on-chips (OOCs) are microphysiological systems/micro-bioreactors that depend on microfluidics to model, not only the microenvironment and architectures of tissues, but importantly also physical stimuli that regulate cell function, such as shear stress introduced via perfusion (Fig. 5) (Park et al, 2012; Cho et al, 2023). Perfusion can also be applied to allow the exchange of nutrients and waste products, or to mimic endocrine signaling. This administration of regulatory molecules in a more physiologically relevant manner can aid in model maturation. For instance, a perfusable endothelial channel in a mammary gland on-chip model was used to apply FGF2 to trigger the branching morphogenesis of an adjacent breast epithelial channel (Kutys et al, 2020). Marshall et al used the perfusability of their microfluidic system to mitigate oxygen diffusion limits and accumulation of waste products, resulting in breast tumor tissue with high viability and more physiologically relevant sizes in vitro than can be obtained in regular, non-perfused cultures (Marshall et al, 2017). In addition, OOC designs can be used to include additional environmental cellular compartments critical for both mammary gland physiology and malignancy, such as adipocytes, fibroblasts, and macrophages, even allowing separate media supplies for these cell types if required (Fig. 5). For instance, BC cells seeded into the lumen of a channel started to invade the surrounding stroma when mammary fibroblasts were introduced into the gel, supporting a critical role for fibroblasts in the transition of BC into an invasive phenotype (Bischel et al, 2015). A connection between cancer-associated fibroblasts (CAFs) and BC invasion could be confirmed using patient-derived primary cells (Truong et al, 2019). Moreover, Lugo-Cintrón et al used a chip model to uncover an underlying mechanism for this interplay, showing that crosstalk between BC cells and CAFs induced the secretion of MMPs, which promoted ECM degradation and, thereby, cancer cell migration into the surrounding tissue (Lugo-Cintrón et al, 2020). Thus, breast-on-a-chip, and especially tumor-on-chip technologies, offer systems that allow for the controlled co-culture of multiple cellular compartments (Fig. 5), shedding light on the role of environmental cells in BC progression. Combination with other tissue engineering techniques, such as biofabrication, could yield more intricate architectural features, such as curvature, demonstrated to be of critical influence on MEC behavior (Nelson et al, 2006) (Fig. 4).

## Biofabrication of breast (cancer)

Biofabrication involves the spatially controlled deposition of cells and biomaterials using additive manufacturing/3D printing methods (Groll et al, 2016). It, thereby, aims to guide cell patterning by generating tissue-like geometrical constraints, cell distributions, and densities (Fig. 5). Various bioprinting techniques, which differ by the types and range of materials that can be used, their resolution, costs, and ease of handling, have been applied to a wide

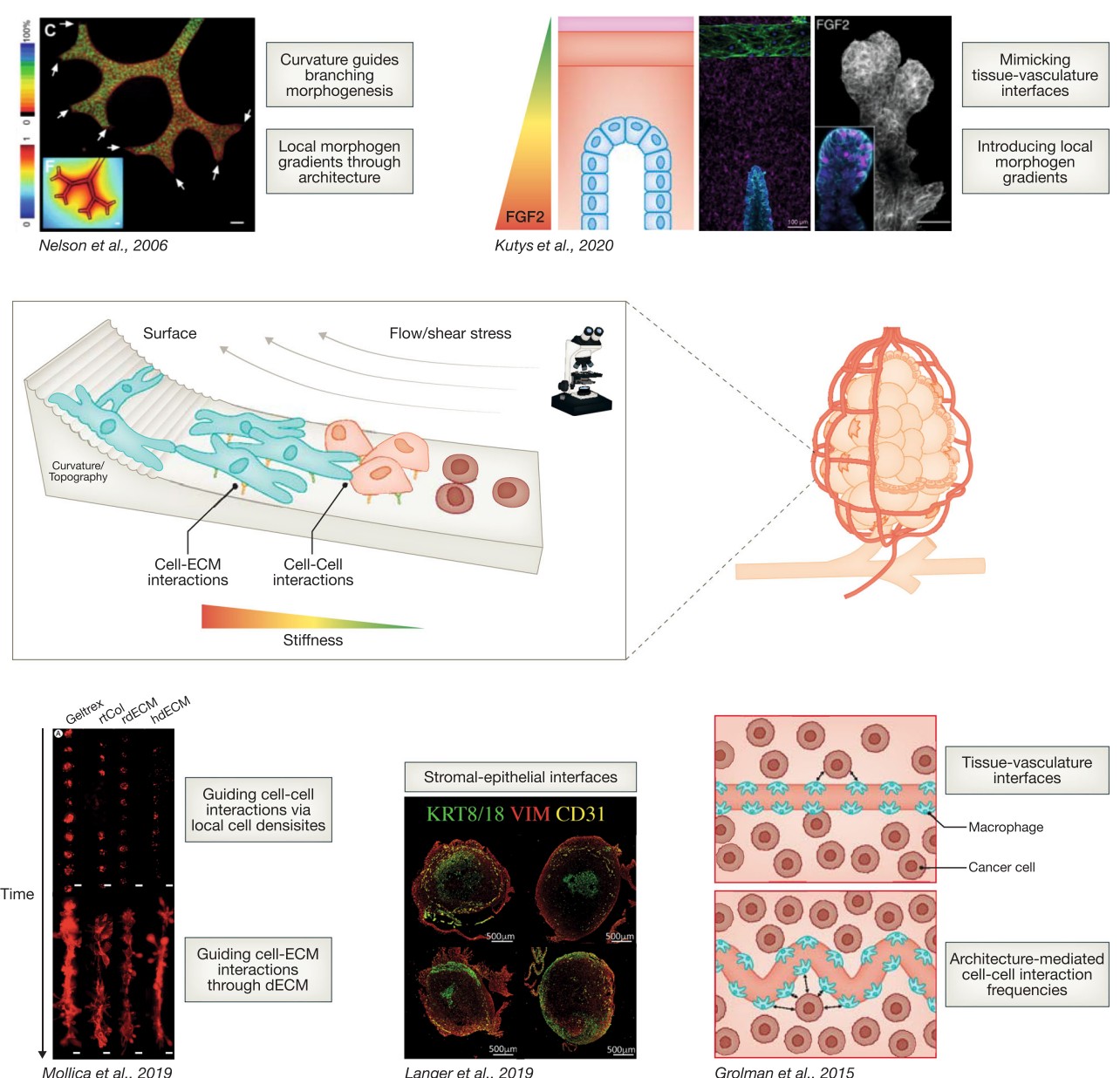

**Figure 4. Approaches tackling different aspects and scales of tissue architecture to guide cell behavior.**

From (Nelson et al, 2006). Reprinted with permission from AAAS. Reproduced with permission from (Kutys et al, 2020) (Creative Commons Attribution 4.0 International License; CC BY 4.0). Reprinted from (Mollica et al, 2019), with permission from Elsevier. Reproduced with permission from (Langer et al, 2019) (Creative Commons Attribution – NonCommercial – NoDerivs (CC BY-NC-ND 4.0).

range of tissues including cardiac tissue (Noor et al, 2019), lung (Grigoryan et al, 2019) and liver (Bernal et al, 2022). Aiming to control the fusion of MECs into large ductal structures, Reid et al used extrusion-bioprinting to deposit breast spheroids and single cells at local cell densities that supported cell aggregation into larger duct-like luminal structures (Reid et al, 2018). Using the same extrusion-bioprinting technique, Mollica et al produced a hydrogel from both human dECM, as well as dECM derived from rat breast tissue and observed the most mature morphogenesis and formation of long ducts in human dECM (Mollica et al, 2019), highlighting again the importance of acknowledging interspecies differences in

ECM composition. While these examples applied extrusion-based technologies, Zhu et al used light-based stereolithography (SLA) bioprinting, applying a photosensitive bioink and selective spatial exposure to light (Fig. 5), to fabricate a BC metastasis model with different pore sizes. By combining bioprinting with hydroxyapatite nanoparticles, an important mineral found in bone tissue, they were able to create a bone-mimetic environment, and BC cells were found to migrate, organize, and form tumor-like structures within this environment (Bahcecioglu et al, 2020; Zhu et al, 2016). Metastasis progression has been shown to be promoted by interactions with macrophages. Therefore, Grolman et al used

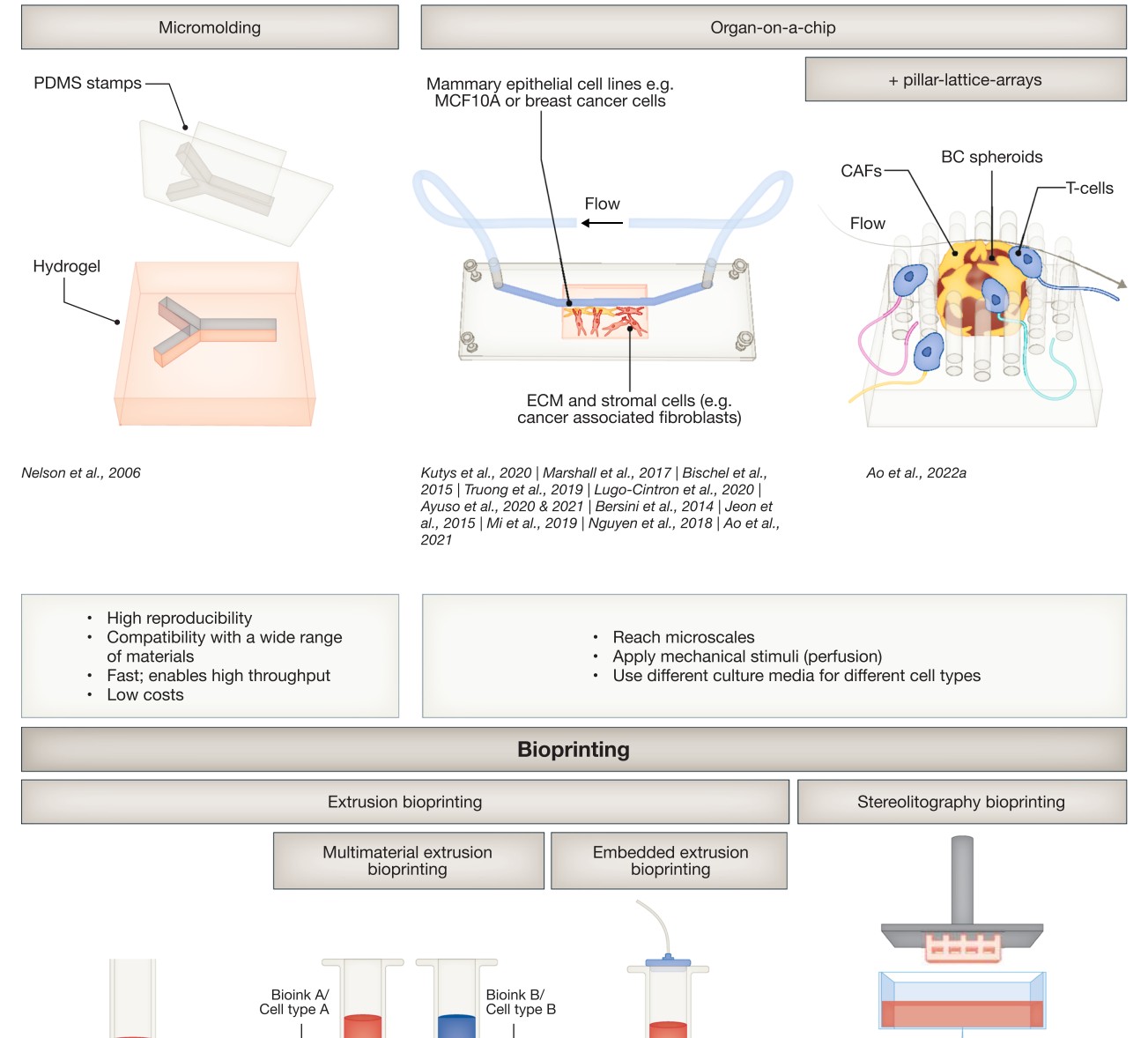

**Figure 5. Different biofabrication approaches addressed in this review.**

Schematic representation of different bioprinting and biofabrication technologies addressed in this review and their key features and advantages.

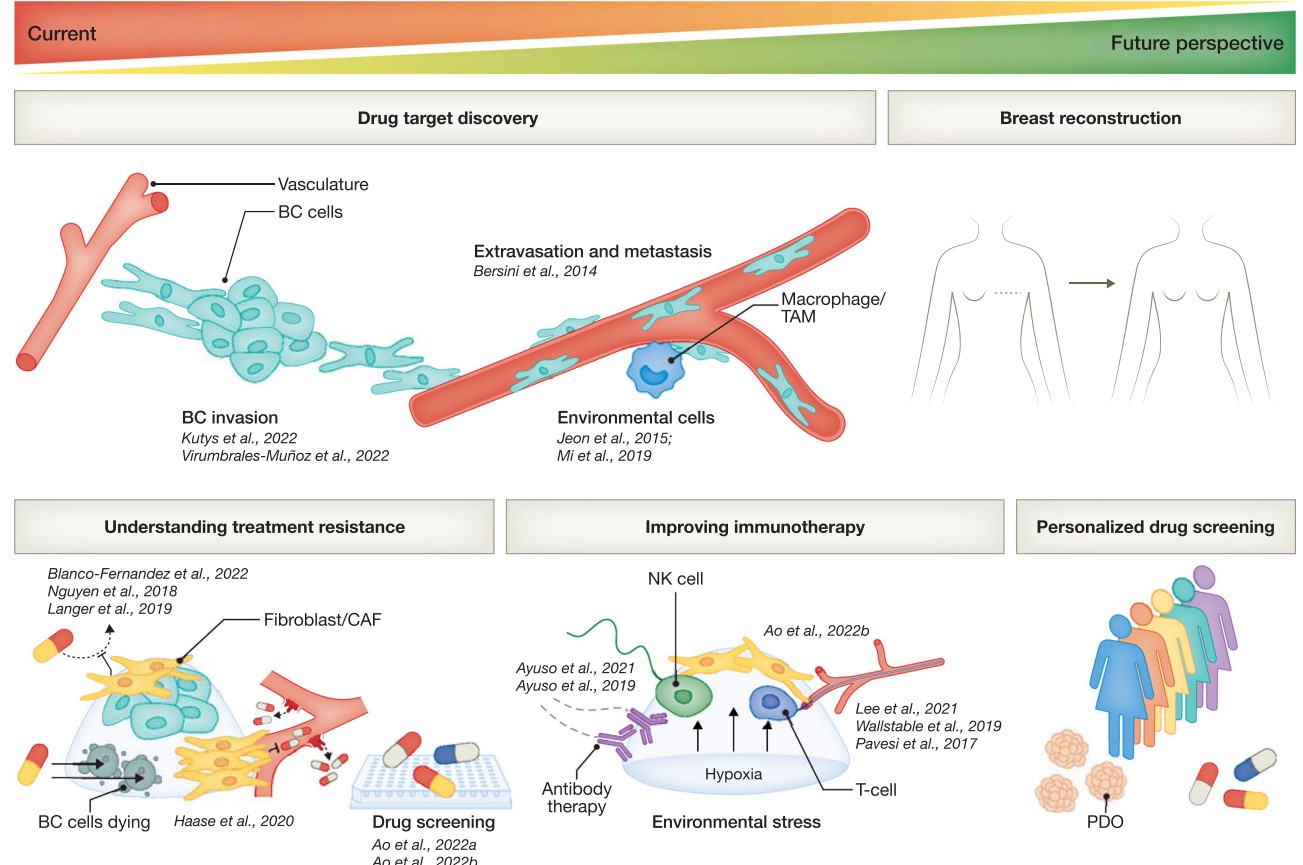

**Figure 6. Current, upcoming, and envisioned future (pre-)clinical applications of breast tissue engineering.**

Application of tissue engineering could lead to the discovery of new drug targets by offering a more complete understanding of BC invasion, extravasation, and metastasis. More architectural complex models can also enhance insight into treatment resistance and pinpoint leads to improved immunotherapy efficacy. Implementation of organoid technology or other patient-representative cell sources could, in the future, enable personalized drug screening. Future clinical application could also be achieved by clinical-grade breast tissue engineering at a sufficient scale for breast reconstruction.

mutlimaterial extrusion-bioprinting to generate vessel-like structures filled with macrophages within a breast adenocarcinoma environment. They exploited this bioprinted model to study the influence of different tissue architectures on macrophage-tumor interactions and found that helically-packed structures enabled more frequent tumor cell-macrophage interactions, prompting the macrophages to enter the tumor tissue and co-localize with the tumor cells (Fig. 4) (Grolman et al, 2015). These promising models make it possible to study the roles of different types of stromal environments in the process of mammary carcinogenesis, however, only rudimentary architectural features of the mammary gland were included that could be further advanced to, for instance, TDLUs.

# Breast tissue engineering for drug development

Applied to BC, breast-on-chip or bioprinted systems can improve our understanding of the biology of disease, leading to the identification of new drug targets, or help to understand and

improve treatment outcomes, including those of increasingly considered immunotherapy (Fig. 6).

## Drug discovery

Kutys et al developed a 3D microfluidic platform that juxtaposes a human mammary duct in proximity to a perfused endothelial channel (Kutys et al, 2020). They used retroviral transfection of MCF-10A cells to model two of the most common genetic alterations underlying the altered ErbB-PI3K-Akt signaling axis prominently occurring in BC; ErbB2 receptor amplification or constitutively active PI3Kα. Both alterations resulted in invasive progression, yet with striking morphological and behavioral differences. This type of insight can shed light on specific targets to treat for BC caused by different genetic alterations and they identified IL-6 as a paracrine factor contributing to vascular dysfunction that specifically occurred with PI3Kα overactivation (Kutys et al, 2020). This also highlights the added value of on-chip-technology for understanding malignancy induced changes in adjacent non-tumor tissue, in this case the vasculature. Ayuso et al used a tumor-lymphatic microfluidic model to investigate gene

 

expression and functional changes in a lymphatic endothelial channel evoked by either ER-positive or triple-negative BC cells (Ayuso et al, 2020). Next to BC invasion (Virumbrales-Muñoz et al, 2019; Kutys et al, 2020), engineered breast models can also be used to study critical steps involved in the highly detrimental and clinically relevant process of tumor metastasis, including extravasation of cancer cells from the vasculature into other tissues (Fig. 6). Bersini et al used a vascularized microfluidic model to study BC extravasation into a bone-mimicking microenvironment, the most frequent site of BC metastasis. They identified CXCR2-CXCL5 as a receptor-ligand pair involved in this process and, thus, as a potential drug target to prevent bone metastasis (Bersini et al, 2014), a finding that was subsequently confirmed by others in an ex vivo culture system (Romero-Moreno et al, 2019). The same group also developed a more advanced bone-extravasation model containing a functional microvascular network embedded in a fibrin gel (as opposed to a single monolayered endothelial channel) and could demonstrate changes in the extravasation behavior of BC cells by preconditioning the micro-vasculature overnight with flow to induce shear stress (Jeon et al, 2015). In addition, they added a macrophage cell line, to model an additional microenvironmental cell type that can affect cancer metastasis. Similarly, Mi et al studied the influence of TAMs on the extravasation of BC cells in a vascularized microfluidic system and demonstrated a bi-directional signaling relationship between TAMs and BC cells; TAMs promoting tumor cell invasion, while BC cells induced the differentiation of this tumor-promoting TAM phenotype (Mi et al, 2019). Further applications of tumor-on-chip technology to study these critical factors involved in metastasis in more detail, could lead to new drug targets to combat this highly unwanted process associated with BC progression and poor prognosis (Cheng and Ueno, 2012).

## Understanding treatment resistance

Tissue engineering can also be applied to study cellular and ECM features contributing to treatment resistance and find ways to overcome them. Cancer stem cells (CSCs) have been recognized as an important cellular player in drug resistance, but for now, the use of tissue engineering to study their specific role has largely been restricted to the application of microfluidics to achieve single-cell seeding (Lamichhane and Tavana, 2024). This strategy was also applied to BC, where single cells were seeded to screen drugs that could reduce their outgrowth into spheroids and, thus, inhibit their clone-forming capacity (Lin et al, 2020). It is expected that tissue-engineered models that accommodate more complex 3D cultures can further advance insight into the unique behavior of these cells and their interaction with drug candidates to delineate specific roles in treatment resistance (Dogan et al, 2021). On-chip technology can also be used to study the importance of interactions between tumor cells and different cellular compartments for treatment outcomes. Nguyen et al used an on-chip 3D co-culture of HER2+ BC cells, fibroblasts, immune, and endothelial cells to examine the response to targeted antibody therapy against the HER2 receptor (trastuzumab). They were able to recapitulate the indirect anti-tumor activity of trastuzumab achieved via antibody-dependent cell-mediated cytotoxicity (ADCC) and show that it relied on the induction of prolonged cancer-immune interactions that were impaired by the presence of CAFs (Nguyen et al, 2018). In a

biofabrication approach, extrusion-bioprinting was used to create a tumor cell core, surrounded by endothelial and stromal cells. Paracrine signaling from fibroblasts was found to influence cancer treatment outcomes, contributing to therapy resistance. This system could be translated to patient-derived tumor cells, highlighting its potential applicability for personalized medicine approaches (Langer et al, 2019). Blanco-Fernandez et al used a similar approach to bioprint BC and stromal cells within a tumor-derived dECM and observed that bioprinted tumor-stromal tissues were less sensitive to doxorubicin treatment than 2D cultured cells or stromal cell-free 3D cultures (Blanco-Fernandez et al, 2022). Moreover, they identified that the addition of collagen-I can reduce sensitivity to this chemotherapy. Another study showed that the geometry of the ECM can impact the responsiveness of breast cancer spheroids to receptor tyrosine kinase (RTK)-targeting drugs (Schwartz et al, 2017). It has also been shown that dense CAFs and collagen-III deposition in the stromal compartment can decrease diffusivity and, thereby, impair drug delivery. Moreover, the same study showed that drug delivery could be severely impacted by tumor-induced vessel dysregulation (Haase et al, 2020). Together, these examples clearly demonstrate the impact of CAFs on BC therapy resistance and further highlight the importance of modeling the immunocompetent TME, as well as the vasculature and ECM interactions, for understanding and predicting treatment outcomes. This strongly applies to immunotherapies, such as already approved checkpoint inhibitors, as well as cellular immunotherapies currently under consideration for breast cancer (Debien et al, 2023), which both can be counteracted by an immunosuppressive TME and depend on the vasculature to reach the tumor site.

## Predicting and improving immunotherapy efficacy

Next to offering controlled culture conditions for a large number of wells for immunotherapy screening and prediction purposes (Ao et al, 2022a), microfluidic models can also be exploited to create additional geometry for studying specific functional features required for immunotherapy efficacy, such as extravasation from the vasculature, homing to and penetration of the tumor by antibodies (Ayuso et al, 2019) and immune cells (Ayuso et al, 2021, 2019; Lee et al, 2021; Ritter et al, 2020; Wallstabe et al, 2019; Maulana et al, 2024), particularly relevant for the targeting of solid tumors (Fig. 6). Ayuso et al used a microfluidic breast tumor-on-chip model to investigate NK cell migration into the tumor, thereby, modeling an important functional requirement of cellular immunotherapy that cannot be captured in simplified culture models. Moreover, perfusion through an endothelial duct combined with high densities of BC cells, resulted in a necrotic tumor core at the distal area from the duct. The resulting hypoxia led to the upregulation of hypoxia-induced genes in NK cells in this area. Recapitulating these microenvironmental stresses occurring in solid tumors, also resulted in the expression of genes associated with exhaustion in NK cells, such as PD1 and CTLA-4. This was accompanied by a reduction in genes involved in cytotoxic function, like Granzyme B. In line with this transcriptomic profile, PD-L1 blocking led to increased numbers of dead tumor cells, indicating that NK cells expressing exhaustion markers are indeed impaired in their cytotoxic function. Importantly, this was not the case in conventional 2D co-cultures of NK and BC cells (Ayuso

 

et al, 2021). Thus, this tumor-on-chip set-up could potentially also be applied to identify new mechanisms involved in functional exhaustion that, in the future, could be targeted to enhance treatment outcomes (Chow et al, 2022). Similar approaches can be applied to model T cell migration and exhaustion and the functional impact of environmental stresses, such as hypoxia, on T cells (Pavesi et al, 2017; Lee et al, 2021; Ritter et al, 2020; Wallstabe et al, 2019; Maulana et al, 2024). For instance, microfluidics and pillar-lattice-arrays were used to instruct the formation of a spheroid tumor core and outer stromal layer consisting of CAFs (Fig. 5). The automated and high-throughput nature of this platform was exploited to screen a drug library and identify a drug that could promote T cell tumor infiltration and treatment efficacy when combined with anti-PD1 immunotherapy (Ao et al, 2022b). This experimental set-up also offers a suggestion of how the application of engineering methods could go beyond physiologically relevant conditions to introduce more extreme geometrical constraints. This could, for instance, be used to select cellular immunotherapy products with enhanced tumor-penetrating capacity by challenging them with a highly dense tumor stroma. In addition, while not a main readout of the study by Ayuso et al, it was observed that the architecture of the collagen hydrogel used impacted NK cell migration, opening opportunities for studying the effect of different ECM compositions on immune cell function (Ayuso et al, 2021). Together, this demonstrates the preclinical potential of BC engineering for performance evaluation and discovery of therapeutic targets to enhance immunotherapy efficacy.

# Organoids as a more representative cell source for breast engineering

While for now predominantly applied with cell lines (Table 2), the combination of breast bioengineering with organoid technology would offer a more heterogenous cell source to create patient-representative models for therapy screening and treatment stratification (Fig. 6) (Ingber, 2022). After earlier work describing 3D breast cultures derived from tissue fragments (Simian et al, 2001; Barcellos-Hoff et al, 1989), MGOs derived from single cells and with long-term culture properties have now been established from different sources (Jamieson et al, 2017; Yuan et al, 2023; Jardé et al, 2016; Fata et al, 2007), including human healthy (Rosenbluth et al, 2020; Dekkers et al, 2020) and breast tumor tissue (Sachs et al, 2018; Dekkers et al, 2021) (Table 2). Mouse MGOs were able to recapitulate key organ-specific architectural and functional features, such as branching morphology, hormone responsiveness, postnatal development, and milk production (Yuan et al, 2023). In addition, BC patient-derived organoids (PDOs) were shown to model patient heterogeneity in hormone receptor status, mutations, and therapy response (Sachs et al, 2018) and are now evaluated in clinical trials for their application to guide personalized treatment decisions (Loewa et al, 2023). Importantly, MGOs are compatible with co-cultures with other cell types, including fibroblasts, which was shown to promote the branching morphogenesis of healthy MGOs (Koledova and Lu, 2017; Sumbal and Koledova, 2019) and the invasive properties of BC organoids (Hanley et al, 2020). In addition, co-cultures of BC PDOs with engineered T cells have been applied to model patient-specific responses to cellular immunotherapy (Dekkers et al, 2023; Maulana et al, 2024). In the near future, tissue engineering approaches could add spatial organization to these organoid co-culture models that is for now lacking. Furthermore, bioengineering can be used to introduce key structural components, such as vascularization and proper cell-ECM interactions (Ascheid et al, 2024; Maulana et al, 2024). Thus, we foresee a clear synergy between organoid technology offering a more representative and diverse cell source and bioengineering to achieve near-physiological tissue complexity in vitro.

# Validating and advancing tissue maturity and functionality

When engineering an advanced in vitro model, it is important to demonstrate that fabricated tissues recapitulate features of the original tissue faithfully and can predict physiological processes and treatment responses. In the next section we will focus on different aspects of mammary gland biology that need to be validated and methods that can be applied to do so.

## Spatially resolved validation

Given the importance of cell organization and the intricate relationship between architecture and function, especially for breast tissue, spatial analysis methods should be considered. MGOs, for example, have been shown to express breast-specific epithelial markers, such as cytokeratin 14 and cytokeratin 8/18, but with limited organization and polarization (Dekkers et al, 2020; Rosenbluth et al, 2020), which is essential for mammary gland function (Roskelley et al, 1995). Spatially resolved technologies will, therefore, be essential, and 3D imaging, particularly multispectral imaging, can provide a comprehensive overview of tissue organization based on molecular markers, cell organization, and cellular interactions. (Ineveld et al, 2022; Rios et al, 2023). In addition, morphological features can be evaluated, such as the presence of polyploid cells reported to play a role in lactation (Rios et al, 2016). Moreover, live cell imaging offers the possibility to assess the contractile properties of the myoepithelial cells, essential for milk release and transport (Sumbal et al, 2020). Spatial transcriptomics can offer a more detailed and unbiased readout of model composition based on gene expression profiles (Wu et al, 2021; Kumar et al, 2023). In addition to these more evident technologies, modeling-based read-outs could also be considered for mammary gland model validation and structural assessment. A recent preprint describes a statistical modeling approach based on a maximum entropy formalism to derive the quantitative relationship between the physical and molecular properties of single cells and the probability distributions of the resulting MGO structures (Srivastava et al, 2023). Mathematical modeling has also been applied to breast models. In their work establishing how curvature guides branching morphogenesis, for instance, Nelson et al mathematically modeled the spacing of ductal units (Nelson et al, 2006). In addition, mathematical models have been used to determine the proliferation and apoptotic dynamics required for proper lumen formation and those associated with ductal carcinoma (Tang et al, 2011), or to identify key processes involved in ductal elongation (Paine et al, 2016). These approaches could also uncover essential parameters of the mammary gland in vivo that in vitro models should adhere to (Paine et al, 2016).

**Table 2. Cell sources applied for in vitro breast modeling.**

| | Model type | Specifics | Origin | Bioengineering application | Ref. |
|---|---|---|---|---|---|
| Cell lines | Non-malignant | MCF-10A | Human mammary epithelium | - Epithelial duct in a vascularized microfluidic platform (Kutys *et al*, 2020)<br>- Non-malignant control for tumor-on-chip models (Ayuso *et al*, 2019; Virumbrales-Muñoz *et al*, 2019; Jeon *et al*, 2015; Mi *et al*, 2019) | (Soule *et al*, 1990) |
| | BC | MCF-7 | Human breast (adeno-carcinoma) | - Vascularized (Ayuso *et al*, 2021, 2019) and lymphatic tumor on chip (Ayuso *et al*, 2020)<br>- Bioprinted co-culture (Langer *et al*, 2019; Blanco-Fernandez *et al*, 2022) | (Soule *et al*, 1973) |
| | | MDA-MB-231-HTB-26 | Human breast (adeno-carcinoma) | - Microfluidic lumen array (Virumbrales-Muñoz *et al*, 2019)<br>- Microfluidic tumor-lymphatic model (Ayuso *et al*, 2020)<br>- Microfluidic 3D tumor model (Wallstabe *et al*, 2019)<br>- Microfluidic co-culture (Mi *et al*, 2019)<br>- Bioprinted co-culture (Langer *et al*, 2019)<br>- Tumor-on-chip (Maulana *et al*, 2024) | (Cailleau *et al*, 1974) |
| | | MDA-MB-231 | Human pleural effusion | Microfluidic bone extravasation model (Bersini *et al*, 2014) | (Yoneda *et al*, 2001) |
| | | MDA-231-BO (bone-seeking clone) | | Micro-vascularized extravasation model (Jeon *et al*, 2015) | |
| | | MDA-MB-468 | | AIM-chip microfluidic device (Ritter *et al*, 2020) | (Cailleau *et al*, 1978) |
| | | BT-474 | Human ductal carcinoma | On-chip co-culture (Nguyen *et al*, 2018) | (Lasfargues *et al*, 1978) |
| | | EO771 | Mouse ductal carcinoma | Mini-tumor chip drug screening (Ao *et al*, 2022a) | (Naour *et al*, 2020) |
| Organoids | Healthy | | Mouse | | (Yuan *et al*, 2023; Jamieson *et al*, 2017; Jardé *et al*, 2016; Fata *et al*, 2007) |
| | | | Human | | (Rosenbluth *et al*, 2020) |
| | BC | | Human | | (Sachs *et al*, 2018) |
| Organoid co-cultures | Healthy | + Fibroblasts | Mouse | | (Koledova & Lu, 2017; Sumbal & Koledova, 2019) |
| | BC | + Fibroblasts | Mouse | | (Hanley *et al*, 2020) |
| | | + T cells | Human | | (Dekkers *et al*, 2023) |
| Organoid-on-chip | Metastatic BC | + CAR T cells | Human | Tumor-on-chip | (Maulana *et al*, 2024) |

*Cellular heterogeneity* (left axis) — *Architectural complexity* (right axis)

## Validation of breast cancer models

While BC models can be validated by comparing tumor heterogeneity, gene signatures, and invasive behavior to the original tumor biopsies, ultimate proof would come from comparing in vitro treatment responses to those observed in patients (Loewa et al, 2023), and the use of PDOs, or other patient-representative cell sources (Voabil et al, 2021), will be key here. Recent work has provided proof-of-concept that a positive hit from a drug screening performed on breast cancer PDOs in vitro, could be confirmed in the clinic, prolonging progression free survival of the individual patient whose PDOs were used to identify the drug candidate (Guillen et al, 2022). Confirming the need for patient-specific models to evaluate therapeutic options, Scabia et al injected patient-derived tumor cells, as well as ER-positive breast cancer cell lines, into murine mammary ducts. They found that, compared to cell lines, response to hormone treatment was generally lower in patient-derived cells and, furthermore, unique for each patient (Scabia et al, 2022). Since treatment administration and outcomes are heavily influenced by both the TME and the vasculature, it will be essential to introduce these critical compartments in further advanced in vitro breast models (Ascheid et al, 2024; Grolman et al, 2015; Langer et al, 2019; Maulana et al, 2024) and characterize and validate their function as well. A thorough analysis of vessel density and barrier function has been executed by Haase et al for a vascularized and perfusable lung carcinoma and ovarian cancer model, demonstrating impaired drug transport in proximity to the tumor. This resulted in reduced treatment efficacy compared to directly treated tumor cells and exemplifies how treatment responses are likely overestimated in models that don't consider the tumor-associated vasculature (Haase et al, 2020). This work, thereby, provides a valuable framework to assess vessel function that could be applied to BC models as well. For example, vessel leakage can be assessed using fluorescently labeled probes, and endothelial and epithelial barrier function, crucial for drug transport, through transepithelial/transendothelial electrical resistance (TEER) measurements (Srinivasan et al, 2015). However, caution is advised when assessing vessel function in vitro as preclinical models have been deemed poor predictors of vessel functionality, exhibiting an increased permeability due to high VEGF production (Dewhirst and Secomb, 2017). Furthermore, a direct comparison of treatment outcomes in vitro to patient responses in the clinic might not always be possible, as patients often receive a multitude of treatments that, together with environmental and nutritional aspects, have an influence on treatment results that cannot be evaluated separately.

## Validation of lactation

Because the ultimate function of mammary gland tissue is breastfeeding, the successful induction of lactation would provide the best evidence of full tissue maturation and functionality. Lactation upon prolactin and oxytocin stimulation has been shown for murine MGOs, demonstrating expression of certain milk proteins and lipid droplets via 3D fluorescent imaging, real-time qPCR, and western blot (Sumbal et al, 2020). Traditionally, the presence of abundant milk proteins, such as caseins, lactalbumin, or lactoferrin is measured by collecting perfused media or cell lysates and performing protein quantifying assays, like Western blotting

and ELISA, to demonstrate lactation (Ballard and Morrow, 2013). However, it has been shown that different milk components require different levels of model maturity and polarization to be produced (Roskelley et al, 1995). Therefore, a more thorough analysis of milk components could deliver more faithful insights into the level of functional maturity. This can be accomplished by analyzing a broader spectrum of milk proteins and the functionally important human milk oligosaccharides (HMOs) (Ballard and Morrow, 2013) via mass spectrometry (Dayon et al, 2021; Lu et al, 2019; Aslebagh et al, 2022; Bao et al, 2013), lipid profiling (George et al, 2023) and analysis of the cellular milk content via single-cell RNA sequencing (scRNAseq) (Twigger et al, 2022). To date, however, lactation has been achieved in murine but not in human breast models, suggesting that cellular organization and functional differentiation is not sufficient yet to enable milk production and raising the question of how this can be reached for these human models.

## Model maturation

Since the functional state of the mammary gland is highly dependent on hormonal signaling and changes drastically during the menstrual cycle and pregnancy (Fig. 2), it is important to ensure spatiotemporal hormone and growth factor supply in mammary gland models to achieve full functionality (Maguire, 2020). While the spatial administration of growth factors leading to directed branching in human MECs has been described in the section on breast-on-chip systems (Kutys et al, 2020), temporal alternation between FGF2 and EGF-containing medium has been shown to have a critical impact on murine MGOs. This resulted in highly complex and branched organoids that more closely resembled native mammary glands (Caruso et al, 2022). In addition, mechanical stimulation, triggered by the newborn's suckling, plays a crucial role in mammary gland functional maturation, both encouraging additional milk production and maintaining lactation. The recent development of a bio-inspired flow simulator provides insight into fluid velocities at different mammary gland locations, offering new opportunities to mimic biologically relevant fluid movements within perfusable mammary gland models (Jiang and Hassanipour, 2021). Combined with this analytic tool, perfusable systems could provide a viable solution for keeping up local and dynamic nutrient and hormone gradients, while also incorporating mechanical stimulation.

# Beyond modeling: biofabrication for breast reconstruction

By 2020, almost 30% of BC patients in the United States underwent either one-sided or double mastectomy to ensure the removal of all tumor tissue or prevent the re-occurrence of BC (Fefferman et al, 2023). Also, in Europa, mastectomy rates have been reported as high as 25% (Garcia-Etienne et al, 2012). However, mastectomy has been associated with lowered body esteem and an increased incidence of psychological disorders, such as depression (Heidari et al, 2015). Therefore, methods to reconstruct breast volume and appearance post mastectomy are in place, including tissue flaps, autologous fat grafting, and synthetic implants (Schmauss et al, 2015). However, these methods are accompanied by complications and limitations. This includes leakage of silicone implants, implant

displacement or deformation, fat tissue volume reduction, donor site morbidity, tissue necrosis, and limited control over final breast shape (Chhaya et al, 2013; O'Halloran et al, 2018). Bioengineered breast tissue could offer an alternative approach that could overcome some of these limitations. Mohseni et al addressed the controllability of the final breast shape, as well as issues in viability associated with big volumes of adipose tissue. They used 3D printing in combination with 3D scanning technology to resemble patient-specific breast shapes pre-mastectomy. An outer scaffold was designed to withstand mechanical forces experienced by the breast in different locations. At the same time, the inner scaffold was developed to contain a gradient of pores and channels to support necrosis-free tissue regeneration and prevent leakage of injected adipose tissue. However, this still needs to be validated using lipoinjection (Mohseni et al, 2019). Chhaya et al developed a method of prevascularizing biofabricated porous scaffolds prior to lipoinjection by implanting these scaffolds in vivo in guinea pigs, resulting in vessel invasion. They could demonstrate adipose tissue regeneration, even in deep layers of their printed constructs. This proves the effectiveness of their prevascularization strategy for maintaining adipose cell viability (Chhaya et al, 2016). While these approaches utilize 3D printing to generate breast scaffolds, due to the injection of adipose tissue after printing, no control over cell placement is achieved. This can be addressed by incorporating cells directly into the printing process. Tytgat et al demonstrated bioprinting of adipose tissue. Their method generated stable constructs with high viability and proliferation of adipocyte-derived stem cells that could be induced to differentiate towards the adipogenic lineage (Tytgat et al, 2019b, 2019a). However, to date, no cell-containing tissue constructs were fabricated at physiologically relevant shape and scale, and biofabrication approaches focus on the restoration of fat tissue volume rather than the functional components of the human breast. New developments in the bioprinting field, like volumetric bioprinting (Bernal et al, 2019), and combinations with other printing methods (Größbacher et al, 2023; Rizzo et al, 2023; Ribezzi et al, 2023) could open possibilities to fabricate large scale tissues. Such methods would allow to incorporate multiple cell types at very short printing times to preserve cell viability. For now, biofabrication of breast tissue at reconstruction relevant scale is still in its infancy. However, we anticipate that developments in printing technology and research on fat, as well as mammary gland tissue engineering, could overcome challenges associated with large-volume fabrication. This offers a promising outlook for the future of functional breast reconstruction (Fig. 6).

## Conclusion and discussion

While playing an important role in human health, many aspects of mammary gland biology are still poorly understood. This calls for more representative models to recapitulate this highly complex and dynamic organ. Bioengineering approaches can contribute to more faithful human mammary gland models. However, there are still challenges laying ahead that need to be tackled to achieve better functional representation. We foresee significant potential in the convergence of organoid technology with bioengineering, due to the increased cellular complexity of organoids and their applicability for personalized medicine. Complexity should also be increased in stromal cellular compartments by the addition of more cell types commonly found in the surrounding of the mammary gland, such as adipocytes, fibroblasts

and endothelial cells. In addition, ECM composition should not be limited to single-material approaches. Although, it has been shown that collagen-I promotes the invasive behavior of MECs, many approaches still depend on the use of collagen-I or similar materials, neglecting the need for a basement membrane for MECs to functionally polarize (Nguyen-Ngoc et al, 2012). While biofabrication technology would be able to deposit multiple materials in different locations, the use of this technology to create interstitial ECM compartments and basement membrane is, for now, underexplored. Similarly, the potential of biofabrication methods, and especially the combination of multiple techniques, to create complex alveolobular architectures has not been fully explored. However, even the most advanced biofabrication techniques will not produce complex constructs that match the level of detail of mature mammary gland tissue. Therefore, post-fabrication maturation strategies will be needed, such as the timed and localized supply of growth factors and hormones, as well as biomechanical stimulation. Finally, bioengineered healthy and cancerous breast models need to be validated from multiple perspectives, with respect to their future applications, and care should be taken that readouts can be brought into spatial context. Here, extensive databases of mammary gland imaging data, spatial transcriptomics, scRNAseq, lipidomics, and proteomics (see also for more information) could be explored to generate a roadmap for validating full mammary gland functionality. Combined efforts in these areas are already cultivating in the preclinical application of breast-engineered models for improving BC treatment (Fig. 6). However, implementation can reach beyond BC, for instance, addressing biological challenges associated with lactation (Roberts et al, 2023; Golan and Assaraf, 2020). In addition, the extent of mammary gland postnatal development, unseen in other organs, has long been recognized (Taylor-Papadimitriou et al, 1977). This unique feature is orchestrated by multiple subpopulations of stem and progenitor cells. Understanding the developmental processes involved in this upkeep of stem cell capacity requires the intact mammary gland niche. A representative, bioengineered model could provide such a niche for studying these stem cells and gaining insight into regenerative processes. Milk-derived stem cells could be exploited as a cell source (Cregan et al, 2007) to circumvent the scarcity of mammary stem cells in non-lactating tissue and the inaccessibility of lactating human breast tissue (Roy et al, 2013). Stem cells from milk have been shown to share similarities to human embryonic stem cells (Hassiotou et al, 2012), and there is data suggesting that these cells can survive the infant's digestive system and integrate into other organs, where they differentiate into functional cells (Hassiotou et al, 2014; Aydın et al, 2018; Twigger et al, 2013). This highlights the importance of breastmilk beyond nutrition and its potential as an accessible cell source for multipotent stem cells. Together, the availability of stem cells from breastmilk and knowledge gained on mammary gland postnatal development, could bring new insight into stem cell biology. Taken together, mammary gland tissue engineering could not only serve as a crucial tool to ensure healthy development of infants and prevent or overcome BC, it could also aid in understanding developmental processes. This knowledge gained on stem cell biology could extrapolate to other organs.

### Pending issues

1. Development of multimaterial approaches that allow to mimic basement membrane and interstitial matrix and that are compatible with biofabrication methods.

2. Incorporation of MGOs as a more diverse and representative cell source into biofabrication processes.

3. Recapitulation of more complex architectural mammary gland features (bifurcations, TDLUs) via the combination of fabrication methods.

4. Implementation of advances in perfusion-bioreactor development, for example, from the field of vascular tissue engineering to better mimic mechanical forces and biochemical gradients found in the breast during lactation (Mitchell et al, 2023), and incorporation of a functional vasculature and lymphatic system.

5. Combining OOCs with bioprinting to increase architectural complexity and development of multi-organ on chips to study metastatic processes (Datta et al, 2020; Cho et al, 2023).

6. Exploit advances in bioprinting technology to reach physiologically relevant large-volume constructs.

7. Combine knowledge gained from breast tissue engineering, vascular tissue engineering, and adipose tissue engineering to recapitulate the mammary gland in its native microenvironment.

8. Combined modeling of key solid tumor features, such as stromal (e.g., CAFs) and immune cell players (e.g. TAMs) alongside structural (e.g., vasculature) and environmental (e.g., hypoxia) factors that can hamper BC treatment efficacy.

## For more information

Datasets of human breast (cancer) tissue and milk:

- Human breast cell atlases: https://navinlabcode.github.io/HumanBreastCellAtlas.github.io/ and E-MTAB-13664, (https://cellxgene.cziscience.com/collections/48259aa8-f168-4bf5-b797-af8e88da6637) (Reed et al, 2024)
- scRNAseq of healthy tissue (GSE161892, GSE180878, GSE113197), BC tissue (GSE182694, GSE161529, GSE176078), and milk (https://www.ebi.ac.uk/biostudies/arrayexpress/studies/E-MTAB-9841, https://www.ebi.ac.uk/biostudies/arrayexpress/studies/E-MTAB-10855, https://www.ebi.ac.uk/biostudies/arrayexpress/studies/E-MTAB-10885)
- Mass spectrometry data on milk (https://glygen.org/global-search-result/Milk) and HMOs (https://glygen.org/publication/PubMed/20578730/)

## Peer review information

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

## Acknowledgements

We would like to thank Jos Malda for valuable feedback on our manuscript. This work was financially supported by the Princess Máxima Center for Pediatric Oncology and Oncode Institute, the Netherlands. RL acknowledges funding from the European Research Council (ERC) under the European Union's Horizon 2020 research and innovation program (grant agreement No. 949806, VOLUME-BIO).

## Author contributions

**Maj-Britt Buchholz**: Conceptualization; Writing—original draft; Writing—review and editing. **Demi I Scheerman**: Writing—original draft. **Riccardo Levato**: Conceptualization; Writing—review and editing. **Ellen J Wehrens**: Writing—original draft; Writing—review and editing. **Anne C Rios**: Conceptualization; Writing—review and editing.

## Disclosure and competing interests statement

M-BB and ACR are listed as inventors on a patent related to breast organoid technology. RL and ACR and M-BB are listed as inventors on a patent concerning biomaterial development. RL is scientific advisor for Readily3D SA, which did not participate in or bias the work.

