## [Peer Review File · EMBO Molecular Medicine]

Human breast tissue engineering in health and disease

Anne Rios, Maj-Britt Buchholz, Demi Scheerman, Riccardo Levato, and Ellen Wehrens

Corresponding author: Anne Rios (a.c.rios@prinsesmaximacentrum.nl)

Review Timeline:

Submission Date:	22nd Mar 24
Editorial Decision:	19th Apr 24
Revision Received:	7th Jun 24
Editorial Decision:	12th Jun 24
Revision Received:	13th Jun 24
Accepted:	14th Jun 24

Editor: Lise Roth

Transaction Report:

19th Apr 2024

Dear Anne,

Thank you for the submission of your review to EMBO Molecular Medicine. We have now received feedback from the experts who agreed to evaluate your manuscript.

As you will see from the reports below, they found the review overall well-written, new and interesting, and are supportive publication pending minor revisions.

We will therefore welcome the submission of a revised manuscript addressing the referees' comments and suggestions at your earliest convenience.

- 1) a .doc formatted version of the manuscript text (including Figure legends and tables)
- 2) Separate figure files
- 3) a letter INCLUDING the reviewer's reports and your detailed responses to their comments.

The manuscript should contain:

1) Glossary: EMBO Molecular Medicine articles are accompanied by a glossary explaining some of the terms used for laymen. Glossary is not a list of abbreviations, abbreviations should be defined the first time they are used in the text

2) For more information: This should be a short list of related web links for further consultation by the readers. (Examples are patient associations, OMIM related links, databases, authors websites, etc.)

3) Pending issues: At the end of each article, there is a box highlighting issues that still need further studies and where research efforts should converge (called the Pending issues box). This should be limited to a few points.

4) Figures:

Please note the following points:

- If there are certain aspects of your figures that are based upon assumptions or where the scientific data remains ambiguous, please add a comment so that we can work with you on an accurate depiction. Please ensure the directionality and nature of interactions is presented accurately.
- If the figure or single panels of the figure have been adapted from a published figure, please add this information to the figure legend (e.g., 'Adapted from...' or 'Based on...').
- Please only re-use figures or parts of a figure if this is essential for understanding the concept communicated. If the figure contains re-used images or elements of images, please make sure that you have the permission/license to publish it (this also applies to your own previous work, if the journal you published in retains copyright. Certain 'creative commons' open access licenses, such as CC-BY 4.0, allow re-use without additional formal permissions). All re-used material must be explicitly cited.
- If you use an image data base for scientific iconography (e.g., BioRender), please let us know if you have a license that allows for publication in an academic journal. Often authors use misleading iconography for expedience. Please ensure the information shown is scientifically accurate.

Looking forward to receiving your revised manuscript,

With my best wishes,

Lise

***** Reviewer's comments *****

Referee #1 (Remarks for Author):

In their review article entitled "Human breast tissue engineering in health and disease", Buchholz et al provide an overview of in vitro models of the healthy and transformed mammary gland, highlighting areas that require further development and proposing strategies for tackling these challenges, with a special focus on materials and tissue engineering. The mammary gland is a fascinating organ - both as a model of developmental, cell and cancer biology, and a tissue of medical importance. It is also one of the organs that was the focus of pioneering efforts in 3D and ex vivo/in vitro culture, with a rich history in technological developments (in addition to biological discoveries). Therefore, the topic of the review is of high fundamental and translational interest and it is also timely. The authors have also done a good job of summarizing the main biological features of the breast, its development and function, and providing a survey of efforts aimed at modeling it in culture, as well as proposing interesting new directions and solutions to challenges. I have several comments, which I hope will be seen as constructive and helpful in further increasing the clarity and completeness of the article.

1. I noticed that, throughout the article, the authors reference studies with both mouse- and human-derived cells (cell lines, organ fragments, organoids etc). This can get confusing. The information is all relevant and insightful, but perhaps the authors can think of a way to structure the manuscript in a way that the reader is aware what model is mouse or human (or what biological concept/feature is mouse vs human).
2. The article extensively focuses on the extracellular matrix, which is interesting, and briefly also mentions the role of adipocytes and fibroblasts in various stages of development, involution etc. The development of the murine gland also appears to be influenced by other types of non-epithelial cells (e.g. macrophages, mast cells, eosinophils - work by Jeffrey Pollard and Zena Werb). Do we have any insights about the role of innate immune cells in the human breast? If so, it would be interesting to mention them, as they (or their factors) could be important in engineered models.
3. Given the focus of the article on the matrix engineering, it would be informative for the authors to include a section detailing the composition and spatial organization of the ECM within the human gland, and how it changes in various stages of development and cancer. Are there any proteomic analyses or multiplexed staining of human breast tissue? How does the ECM of the human gland differ (or not) from the mouse gland (e.g. the characteristic collagen deposition at the stalks of the branches, the basement membrane thinning at TEBs and sites of lateral branching etc).
4. Related to the previous point, this reviewer didn't quite understand the rationale behind the sections focused on bioprinting and biofabrication, especially given that relatively simple gel mixes (Sokol et al, 2016) gave rise to beautiful human mammary structures in vitro, which underwent branching, were responsive to hormonal stimulation etc. If there is a specific limitation that was not addressed by the mixed-ECM gels and which motivates the need for biofabrication, the authors should introduce it more clearly.
5. Similarly, what are the main challenges solved or advantages provided by geometric constraints or on-chip systems (I am sure there are many, but the authors should briefly introduce them in relation to breast models, at the beginning of the relevant section).
6. The Validation of breast cancer models section is a little underdeveloped. I think the problem is that the question/problem of breast cancer model validation is so large and complex, that a simple paragraph does not do it justice. The authors could try adding more details on what has been done, what is out of reach and why. Or they could provide several examples of drug resistance and discuss how breast models could be developed to capture and investigate these cases. Alternatively, if discussing the topic in a meaningful way would require too much additional space, time and effort, they could simply remove this section.
7. In the section on organoids, I believe that the historic work of Mina Bissell on 3D models should be recognized. They even used the term 'organoid' to (accurately) describe mammary gland fragments growing and branching in 3D gels, before Hans Clevers' group popularized the approach in the later 2000s (e.g. Barcellos-Hoff et al, 1989; Simian et al, 2001; Fata et al, 2007).
8. Minor: there are some typos in the manuscript (e.g. 24, 133, 223, 560).

Referee #2 (Remarks for Author):

Buchholz et al. present a review titled "Human breast tissue engineering in health and disease." This review article provides a high-level survey of mammary gland development, composition, and function and then goes on to detail various engineering approaches to modeling in the culture setting. Strengths of the work include its breadth; describing the structure of the gland at

the cellular and ECM level, juxtaposed to different engineered, decellularized, and organoid approaches; describing their benefits, challenges and limitations, as well as applications to drug screening and predicting immunotherapy response. Weaknesses come from the breadth of the review -- leading to sections that are sometimes superficially discussed (discussed in comments below) and the existence of reviews with partially overlapping content. Despite these weaknesses, this is a well written review, generally clear and informative, and would be of benefit to the community.

Reviewer comments:

Abstract line 24 "fysiology"; Introduction lines 43-48 would benefit from slight rewording and punctuation changes to facilitate clarity and ease of reading.

"Intricate interplay of cell composition, tissue architecture and hormonal regulation" (starting line 69): This section gives a very high-level overview and would benefit from some additional depth. For example, growth factor regulation of gland development at puberty provides a framework to highlight the complexities of postnatal regulation and could be expanded upon starting around line 85. Estrogen-regulated induction of amphiregulin and interaction this growth factor with epithelial and stroma compartments is required for gland development. This example, and/or others like it, could be used to briefly but more compellingly foreshadow challenges facing complex engineered model generation. This should be taken as a recommended but non-essential opinion of the reviewer, with the aim of making this already very nice review stronger.

Table 2: The authors describe MCF10 cells as "healthy." This cell type should be referred to as non-malignant. It gives rise to cells with "normal" appearing morphology but the line itself is from fibrocystic breast tissue, near diploid but not "normal" and is p16 null.

The authors do not discuss efforts to mathematically/physically understand tissue architecture and signaling, or tissue engineering combined with computational modeling in a live-cell setting. These efforts go hand-in-hand with the engineering approaches discussed herein, 1) because they rely on these very engineering approaches to conduction biological/ mechanistic studies, and 2) they inform iterative improvement of engineered models to better recapitulate in vivo structure/function. Some topic examples for consideration include:

<https://pubmed.ncbi.nlm.nih.gov/34965432/>

<https://pubmed.ncbi.nlm.nih.gov/36113484/>

<https://pubmed.ncbi.nlm.nih.gov/37425903/> (selected example of emerging studies)

Referee #3 (Remarks for Author):

In the present review, the authors set the stage by discussing the biological significance of the mammary gland, its implications in breast cancer, and the limitations of current research models. They argue for the necessity of more complex and representative in vitro models to enhance understanding and treatment of breast health and pathology.

The authors summarized current landscape of breast tissue engineering, including various approaches such as organoids, decellularized extracellular matrices, use of biomaterials, and biofabrication techniques (SLA, 3D bioprinting, etc) to mimic the native mammary gland environment. The authors also outlined requirements and challenges in creating predictive human breast models.

Next, the authors summarized potential applications of engineered breast tissues in preclinical studies, especially for improving understanding and treatment of breast cancer, and discussed the usage of human-derived materials in tissue engineering, and discussed the translational potential of these technologies into clinical practice, including therapeutic applications and personalized medicine.

In the discussion, the authors highlighted the need for more complex models that integrate intricate tissue architectures and cellular interactions, and discussed the potential of combining tissue engineering with other emerging technologies to enhance model fidelity and functionality.

Overall impression

The manuscript is well-structured, informative, and timely in the field of tissue engineering and mammary gland biology. It thoroughly discusses the current status, advancements, and challenges in creating representative models of human breast tissue, which are crucial for understanding disease mechanisms and for the development of targeted therapies. The integration of multiple disciplines, including cellular biology, materials science, and bioengineering, is a notable strength of the review.

Strengths

- Comprehensive coverage of the biological and engineering aspects relevant to breast tissue engineering for both normal and malignant tissue.
- Inclusion of a wide range of references.
- Detailed review on the application of organ-on-chip and bioprinting technologies in breast tissue modeling.
- Specifically, Figure 3 is an efficient way for visualizing and comparing the different features and applications from multiple models.

Shortcomings are discussed below as comments

Line 69: Not only architecture is a limitation in organoid, but also "organoid-to-organoid" reproducibility in phenotype and cellular complexity, which hinders its application in drug screening. Reference: section "Towards a new generation of organoids". G. Rossi, A. Manfrin, M. Lutolf. Nature Reviews Genetics, 2018

Line 71: TEBS and TDLU are not the same. TEBS are transient structures found in the pubertal gland while the ducts elongate and may also be found in the pubertal breast in humans, see studies of the human breast by Barry

Line 73f: myoepithelial cells should not be equated to basal cells. The former are defined based on the expression of contractile proteins the latter are defined by their position and comprise the former.

line 80: better say "human breast" and "mouse mammary gland".

Line 88: depending on the mouse strain with repeated estrous cycles some side branching or tertiary branching can be observed

Lines 84, 94, 110: lacking references

The term "lobuloalveolar" is wrong. Lobe or lobules imply a separation within the tissue by sheets of connective tissue that do not exist in the mouse mammary gland. The human breast has lobes.

Lines 104 ff "This basement membrane, produced by myoepithelial cells and stromal fibroblasts, is a key regulator of the integrin-mediated polarized mammary epithelial phenotype."

This reviewer is under the impression that the epithelial cells also contribute to the basal lamina. Please, provide reference.

Line 146 floating collagen gels: the original work by Joan Emmerman should be quoted here.

Lines 175 ff However, because of interspecies differences, human tissue is needed to faithfully recapitulate the human mammary gland ECM, which is largely restricted by availability and ethical constraints."

This statement is misleading. Reduction mammoplasty tissue can be readily obtained under ethic protocols and indeed their use has lead to important lines of work starting with Magaret Stampfer.

325 ff Section on drug discovery. Here interesting findings in different models are listed but it is not mentioned to what extent these were validated clinically. In other words is there any evidence that findings in these models can be extrapolated to in vivo settings?

Introduce the engineering terms such as "extrusion-bioprinting", "pillar-lattice-arrays". Indeed, given the length of the review a schematic overview and comparison of the approaches that have been used to date will be helpful.

Line 356: In the section of "Understanding treatment resistance", the authors reviewed several works on the role of CAFs in rendering drug resistance to breast cancer in on-chip systems. However, this section would be broadened and more thorough if the authors can include works investigating other factors contributing to drug resistance on-chip, such as Cancer stem cells (CSCs), ECM remodelling, EMT, etc.

Line 391-393 sentence unclear

Line 453: The authors could include the work from Alana Welm's laboratory, Nature Cancer, 2023, as another example of validation of BC model largely for triple negative breast cancers

"While BC models can be validated by comparing tumor heterogeneity, gene signatures and invasive behavior to the original tumor biopsies, ultimate proof would come from comparing in vitro treatment responses to those observed in patients (Loewa et al., 2023), and the use of PDOs, or other representative cell sources, will be key here."

And the work with intraductal xenografts Scabia et al Nat Com 2022.

Line 482 "endocrine hormonal" use one or the other

Puberty and menstrual cycles should be shown as separate phases.

Line 499 By 2020, almost 30% of BC patients underwent either one-sided or double mastectomy to ensure removal of all tumor tissue, or prevent re-occurrence of BC (Fefferman et al., 2023).

Are you sure??? This sounds wild.

The last section, as it is, does not add to the review. Materials that are introduced into the human body need to fulfill entirely different requirement than the different models and deserve a separate review.

Problem with MDA MD 231 cells

Balance in Literature Presentation:

The literature review is thorough, but the authors could also show some conflicting findings or controversies within the field to provide a more balanced view.

More importantly, the authors should share more their points of view and including comparison, criticism, future perspectives, etc. Because "Writing a review article is about more than seizing on an interesting topic and gathering the relevant references, ... readers should learn something(s) that they couldn't get just by reading the references. Therefore, make sure you include your point(s) of view": <https://www.elsevier.com/connect/six-things-you-should-keep-in-mind-when-writing-a-review-article>

Summary tables and timelines:

Additional summary tables comparing the features of different model systems discussed could provide a quick reference for readers.

Timelines summarizing the development of the main technologies or milestones in the field and their applications would enhance understanding and engagement.

Readability:

The review is generally straightforward and comprehensible albeit somewhat lengthy. The presence of long and fragmented sentences can disrupt the reading flow, resulting in an uneven reading experience.

For example, certain sentences could be clarified to enhance understanding: When they modeled two of the most common genetic alterations underlying the altered ErbB-PI3K-Akt signaling axis prominently occurring in BC; (new sentence started)

The writing can be more efficient and concise while including more information for readers to comprehend and picture the literatures. For example, in line 390-391, description for hypoxia can be shortened by eliminating expressions of "microenvironmental stresses arising at the solid tumor core"; and in line 391-392 "In addition, their chip design could to some extent model NK cell exhaustion, which was not the case...", it would be more informative to also describe how the modeling of NK cell exhaustion is done, to what extent and why there seem to be a limitation. Similar lack of cleanness and elaboration are occasionally found in other parts of the review.

Dear Anne,

Thank you for the submission of your review to EMBO Molecular Medicine. We have now received feedback from the experts who agreed to evaluate your manuscript. As you will see from the reports below, they found the review overall well-written, new and interesting, and are supportive publication pending minor revisions.

We will therefore welcome the submission of a revised manuscript addressing the referees' comments and suggestions at your earliest convenience.

We are very pleased with the overall positive evaluation of our work by the reviewers and their unanimous opinion that our review is of interest and publishable. We have now addressed all the minor revisions requested by the reviewers in the revised manuscript, as well as in the point-by-point reply to the reviewers' comments below. Specifically, our manuscript has been improved in scope and readability and now includes new **Figure 5** providing a schematic summary of the main tissue engineering technologies discussed and their key advantages.

- 1) a .doc formatted version of the manuscript text (including Figure legends and tables)
- 2) Separate figure files
- 3) a letter INCLUDING the reviewer's reports and your detailed responses to their comments.

The manuscript should contain:

- 1) Glossary: EMBO Molecular Medicine articles are accompanied by a glossary explaining some of the terms used for laymen. Glossary is not a list of abbreviations, abbreviations should be defined the first time they are used in the text

We have now edited the glossary to offer a list of definitions and laymen explanations of the common engineering and scientific terms used in our manuscript.

- 2) For more information: This should be a short list of related web links for further consultation by the readers. (Examples are patient associations, OMIM related links, databases, authors websites, etc.)
- 3) Pending issues: At the end of each article, there is a box highlighting issues that still need further studies and where research efforts should converge (called the Pending issues box). This should be limited to a few points.

4) Figures:

Please note the following points:

- If there are certain aspects of your figures that are based upon assumptions or where the scientific data remains ambiguous, please add a comment so that we can work with you on an accurate depiction. Please ensure the directionality and nature of interactions is presented accurately.
- If the figure or single panels of the figure have been adapted from a published figure, please add this information to the figure legend (e.g., 'Adapted from...' or 'Based on...').
- Please only re-use figures or parts of a figure if this is essential for understanding the concept communicated. If the figure contains re-used images or elements of images, please make sure that you have the permission/license to publish it (this also applies to your own previous work, if the journal you published in retains copyright. Certain 'creative commons' open access licenses, such as CC-BY 4.0, allow re-use without additional formal permissions). All re-used material must be explicitly cited.

We carefully checked whether the images used in these figures were covered under an open access license or obtained the formal permission to re-use them. In the corresponding figure legends, we added the required information to adhere to the terms and conditions for re-use.

- If you use an image data base for scientific iconography (e.g., BioRender), please let us know if you have a license that allows for publication in an academic journal. Often authors use misleading iconography for expedience. Please ensure the information shown is scientifically accurate.

We did not rely on an image database and created all the schematics used in the figures ourselves.

Looking forward to receiving your revised manuscript,

With my best wishes,

Lise

***** Reviewer's comments *****

Referee #1 (Remarks for Author):

In their review article entitled "Human breast tissue engineering in health and disease", Buchholz et al provide an overview of in vitro models of the healthy and transformed mammary gland, highlighting areas that require further development and proposing strategies for tackling these challenges, with a special focus on materials and tissue engineering. The mammary gland is a fascinating organ - both as a model of developmental, cell and cancer biology, and a tissue of medical importance. It is also one of the organs that was the focus of pioneering efforts in 3D and ex vivo/in vitro culture, with a rich history in technological developments (in addition to biological discoveries). Therefore, the topic of the review is of high fundamental and translational interest and it is also timely. The authors have also done a good job of summarizing the main biological features of the breast, its development and function, and providing a survey of efforts aimed at modeling it in culture, as well as proposing interesting new directions and solutions to challenges. I have several comments, which I hope will be seen as constructive and helpful in further increasing the clarity and completeness of the article.

We are very pleased that the reviewer finds the topic of our review timely and of high fundamental and translational interest. We thank the reviewer for the careful evaluation and the constructive feedback offered that has helped to further shape our manuscript and improve clarity.

1. I noticed that, throughout the article, the authors reference studies with both mouse- and human-derived cells (cell lines, organ fragments, organoids etc). This can get confusing. The information is all relevant and insightful, but perhaps the authors can think of a way to structure the manuscript in a way that the reader is aware what model is mouse or human (or what biological concept/feature is mouse vs human).

We understand this notion and already entertained alternative writing structures when drafting the first version of our manuscript that would allow for better discrimination between mouse and human data. However, a structure guided by main biological and engineering topics, rather than mouse versus human, was better justified given the overall scope of our review on breast tissue engineering to capture *human* breast function. We did try to be very transparent throughout the manuscript text about whether a certain study described findings, or a model, based on mouse or human cells or organoids. Also, by consistently using the terms *mammary gland*, when referring to mouse and *breast*, when referring to human. In addition, we included in both **Table 1** and **Table 2** the species origin of the cell sources used. Finally, in line with the reviewer's 3rd comment, in our section on mammary gland ECM we have now tried to be more transparent on what has been shown for human breast ECM composition and function, as compared to more extensive analysis performed in mouse models (see a newly introduced sub-section on page 7 of our revised manuscript and lines 234-238 on page 10).

2.The article extensively focuses on the extracellular matrix, which is interesting, and briefly

also mentions the role of adipocytes and fibroblasts in various stages of development, involution etc. The development of the murine gland also appears to be influenced by other types of non-epithelial cells (e.g. macrophages, mast cells, eosinophils - work by Jeffrey Pollard and Zena Werb). Do we have any insights about the role of innate immune cells in the human breast? If so, it would be interesting to mention them, as they (or their factors) could be important in engineered models.

Our description of stromal cellular players (and the need to model them) was largely focused on fibroblast and macrophages, as many breast engineering strategies have been aimed at uncovering the role of cancer-associated fibroblast (CAFs) or tumor-associated macrophages (TAMs) in breast cancer progression and treatment response. See the work we cover in the section on *breast tissue engineering for drug development*, starting on page 17. While we do not have space to go into extensive detail, we agree with the reviewer that that cuts short essential work performed by leading scientists in the field, like Jeff Pollard and Zena Werb. Therefore, in our introductory section describing the interplay between cell composition, architecture and dynamic regulation of the mammary gland, we now refer to their work, highlighting the role of recruited immune cells in postnatal mammary gland development (macrophages in lines 108-110 on page 5; and mast cells and eosinophils in lines 124-126 on page 6). In addition, we added the role of tissue-resident ductal macrophages in mammary gland homeostasis (lines 107-108, page 5) and involution (line 132, page 6). We also included additional references to stress the multifaceted role of fibroblasts in mammary gland developmental stages (lines 104-107 on page 5) and adipocytes during involution (Zwick *et al.*, 2018 lines 134-135 on page 6). In addition, we now list these cell types on page 15, lines 367-368 of our revised manuscript as essential cellular compartments that can be introduced through organ-on-chip technology. We hope that with these additions, we now offer a more balanced overview of the stromal and immune cells at play in both normal mammary gland development and cancer and the critical need to include them through advanced engineering strategies.

3. Given the focus of the article on the matrix engineering, it would be informative for the authors to include a section detailing the composition and spatial organization of the ECM within the human gland, and how it changes in various stages of development and cancer. Are there any proteomic analyses or multiplexed staining of human breast tissue? How does the ECM of the human gland differ (or not) from the mouse gland (e.g. the characteristic collagen deposition at the stalks of the branches, the basement membrane thinning at TEBs and sites of lateral branching etc).

We followed the reviewer's suggestion and now on page 7 of our revised manuscript introduce a new sub-section that outlines knowledge on breast ECM composition and function that has been obtained from human data. We included a notion here that this is still limited, as compared to what has been analyzed in mice, for instance, the dynamic changes in ECM composition occurring during puberty that the reviewer is referring to and that we now included on page 8, lines 175-181 of our revised manuscript. Thus, given the importance of ECM in dictating cell behavior and considerable interspecies differences in its

composition, as noted in lines 158-162 on page 7 of our revised manuscript, more detailed data on human breast ECM composition is warranted. We acknowledge this in the concluding paragraph of the next section on guiding cell organization through biomaterial modulation (lines 234-238 on page 10).

4. Related to the previous point, this reviewer didn't quite understand the rationale behind the sections focused on bioprinting and biofabrication, especially given that relatively simple gel mixes (Sokol et al, 2016) gave rise to beautiful human mammary structures in vitro, which underwent branching, were responsive to hormonal stimulation etc. If there is a specific limitation that was not addressed by the mixed-ECM gels and which motivates the need for biofabrication, the authors should introduce it more clearly.

We agree with the reviewer about the importance and the implementation of the work performed by Sokol *et al.*, which is also why we highlight their results quite prominently in our review. Their work clearly demonstrates the importance of incorporating a multi-material ECM composition, supporting various proteins, as well as a carbohydrate component. However, even in these mixed hydrogels the frequency of complex geometries is quite low. Taking also into consideration seeding efficiency, this is only observed in 22% of the cultures, when starting with cell clusters from mammary reduction material. In the case of single cells as a starting point, this frequency dramatically drops below 0.0075%. Moreover, the few structures with complex morphologies that do develop from single cells only contained cytokeratin 14 (CK14)-positive basal cells and no CK8/18 expressing luminal cells. So, not only proper polarization highly essential to mammary gland function was lacking, one of its key cellular subtypes did not differentiate using this approach. Therefore, we strongly feel that adding relevant tissue constrains through engineering technologies could still offer profound advantages when combined with a multi-material ECM. Especially because of evidence that induced architecture can be sufficient to stir cells towards functional differentiation, shown by the work from Gjorevski *et al.* (*Science*, 2022). Furthermore, biofabrication technologies can help to achieve full model maturation with additional and near physiological functionalities (extending beyond hormone-responsiveness demonstrated by Sokol *et al.*). For instance, next to introducing additional cell types, perfusable organ-on-chip systems could offer a means to provide a timely sequence of hormone supply, as well as mechanical stimulation that will be important for the induction of lactation. Precise control over tissue architecture, for instance through bioprinting, can also offer experimental flexibility and adaptability to ensure model accessibility to specific read-outs or research questions aimed to address. To put forward this vision on the added value of biofabrication techniques in the revised manuscript more clearly, we made several adaptations.

- Where we describe the work of Sokol *et al.* we added a notion on the frequency, and thus reproducibility, of complex ductal and lobular morphologies observed with their mixed hydrogel approach (lines 228-230, page 10)
- In our section on *capturing organ architectures*, we added a sentence to not only highlight how architecture can have a determining role in tissue function, but also

how experimental control over architecture can aid for reproducibility and model accessibility (lines 349-351, page 15).

- In our sub-section on *breast-on-chip* systems, we added that the perfusability of these systems can aid in model maturation via controlled hormone supply (lines 359-360, page 15). In addition, we further expanded on the advantages of these systems for the combined culture of various cellular compartments (lines 367-368, page 15). This is in addition to the spatiotemporal hormone and growth factor supply and mechanical stimulation that we already describe in our sub-section on *model maturation* that will be critical for full functionality.
- We added new **Figure 5** that offers both a schematic representation of the different bioprinting and organ-on-chip methods discussed, as well as describes the main advantages of these technologies.

We hope that with these additions, our vision of why biofabrication is important, in addition to TME engineering, is now more clearly presented in the revised manuscript.

5. Similarly, what are the main challenges solved or advantages provided by geometric constraints or on-chip systems (I am sure there are many, but the authors should briefly introduce them in relation to breast models, at the beginning of the relevant section).

In addition to the other changes described in more detail above, we now expanded on this in the introductory paragraph of *breast-on-chip systems* (lines 359-360, page 15 and lines 367-368, page 15) and *biofabrication* (lines 382-384, page 16). In the continuation of the latter section and the following section on *drug development*, we furthermore offer clear examples of the added value of introducing tissue-like geometrical constraints and cell distributions and densities through bioprinting. E.g. having the right cellular density can promote the formation of duct-like structures (Reid *et al.*, 2018) (lines 386-389, page 16). Moreover, the work of Ao *et al.* (2002) (lines 502-506, page 20) suggest that implementation of geometrical constraints could even expand beyond modelling what is physiologically relevant, as an extra dense stroma surrounding the tumor could, for instance, allow to select cells for cellular immunotherapy treatment with enhanced tumor penetrating properties (lines 506-510, page 20/21). We hope that with these introductions and examples, as well as the ones described in our response to the reviewer's previous comments, the added value and future perspective of tissue engineering is clearly outlined in our revised manuscript.

6. The Validation of breast cancer models section is a little underdeveloped. I think the problem is that the question/problem of breast cancer model validation is so large and complex, that a simple paragraph does not do it justice. The authors could try adding more details on what has been done, what is out of reach and why. Or they could provide several examples of drug resistance and discuss how breast models could be developed to capture and investigate these cases. Alternatively, if discussing the topic in a meaningful way would require too much additional space, time and effort, they could simply remove this section

We apologize for being a bit short in this section in the first version of our manuscript, due to space constraints. We have now included more depth, citing 4 additional recent papers. This

work highlights what has been done already in terms of validation of breast cancer models (e.g. Guillen *et al.* 2022 and Scabia *et al.* 2022; lines 578-584, page 25). In addition, we outline what still needs to be achieved in terms of establishing further advanced *in vitro* models and how those should be validated regarding the TME and, especially, newly introduced vasculature (lines 584-596, page 25). Finally, we discuss the expected challenges that could be encountered with these validation efforts (lines 596-602, page 25/26).

7. In the section on organoids, I believe that the historic work of Mina Bissell on 3D models should be recognized. They even used the term 'organoid' to (accurately) describe mammary gland fragments growing and branching in 3D gels, before Hans Clevers' group popularized the approach in the later 2000s (e.g. Barcellos-Hoff *et al.*, 1989; Simian *et al.*, 2001; Fata *et al.*, 2007).

While some of the requirements that are now part of the broadly used definition of organoids were not assessed, e.g. derived from single (stem) cells (not structures) and exhibiting the capacity of long-term propagation in culture, we agree that these early 3D breast cultures are worth to include. We now mention these efforts in the revised version of our manuscript (lines 520-522 on page 21).

8. Minor: there are some typos in the manuscript (e.g. 24, 133, 223, 560).

We apologize for this oversight. We have now corrected the specific typos mentioned by the reviewer and carefully checked the entire manuscript again for additional typos and corrected them.

Referee #2 (Remarks for Author):

Buchholz *et al.* present a review titled "Human breast tissue engineering in health and disease." This review article provides a high-level survey of mammary gland development, composition, and function and then goes on to detail various engineering approaches to modeling in the culture setting. Strengths of the work include its breadth; describing the structure of the gland at the cellular and ECM level, juxtaposed to different engineered, decellularized, and organoid approaches; describing their benefits, challenges and limitations, as well as applications to drug screening and predicting immunotherapy response. Weaknesses come from the breadth of the review -- leading to sections that are sometimes superficially discussed (discussed in comments below) and the existence of reviews with partially overlapping content. Despite these weaknesses, this is a well written review, generally clear and informative, and would be of benefit to the community.

We are very pleased that the reviewer considers are work of high quality and beneficial to the scientific community. We are grateful for the insightful feedback offered by the reviewer that has helped us to increase the depth and focus of our review.

Reviewer comments:

Abstract line 24 "physiology"; Introduction lines 43-48 would benefit from slight rewording and punctuation changes to facilitate clarity and ease of reading.

We edited this for the specific lines pointed out by the reviewer (lines 64-69 on page 4 of our revised manuscript). In addition, we carefully went through the entire manuscript again to identify instances where we could enhance clarity and readability and improved this on several occasions (e.g. lines 413-418 on page 17 and lines 650-657 on page 27/28).

"Intricate interplay of cell composition, tissue architecture and hormonal regulation" (starting line 69): This section gives a very high-level overview and would benefit from some additional depth. For example, growth factor regulation of gland development at puberty provides a framework to highlight the complexities of postnatal regulation and could be expanded upon starting around line 85. Estrogen-regulated induction of amphiregulin and interaction this growth factor with epithelial and stroma compartments is required for gland development. This example, and/or others like it, could be used to briefly but more compellingly foreshadow challenges facing complex engineered model generation. This should be taken as a recommended but non-essential opinion of the reviewer, with the aim of making this already very nice review stronger.

We agree with the reviewer that very brief examples can already help to underscore the complexity of mammary gland regulation and the uniqueness of this organ in terms of postnatal development. Therefore, we followed the reviewer's suggestion and now provide more detail on pubertal morphogenesis, describing the role of recruited immune cells in this process (lines 124-126 on page 6). In addition, we now reflect puberty as a separate developmental stage in **Figure 2**. We also acknowledge in lines 111-112 on page 5 that growth factors produced by the stroma, next to hormones provided by the pituitary gland and ovaries, all play a role in postnatal regulation. Finally, we provide a clear example of a complex interaction between multiple cell types and factors playing a role in mammary gland structural development, involving recruited immune cells, stromal cells and mammary gland epithelial cells and their secreted factors, as well as hormonal regulation (lines 113-118 on page 5/6). Altogether, this creates a more comprehensive picture of the complexity of mammary gland tissue regulation that developed model system should ideally aim to adhere to.

Table 2: The authors describe MCF10 cells as "healthy." This cell type should be referred to as non-malignant. It gives rise to cells with "normal" appearing morphology but the line itself is from fibrocystic breast tissue, near diploid but not "normal" and is p16 null.

We thank the reviewer for pointing out this nuance and have now changed to non-malignant where we refer to this cell line in **Table 2** on page 23 of our revised manuscript.

The authors do not discuss efforts to mathematically/physically understand tissue architecture and signaling, or tissue engineering combined with computational modeling in a live-cell setting. These efforts go hand-in-hand with the engineering approaches discussed herein, 1) because they rely on these very engineering approaches to conduct biological/mechanistic studies, and 2) they inform iterative improvement of engineered models to better recapitulate in vivo structure/function. Some topic examples for consideration include:

<https://pubmed.ncbi.nlm.nih.gov/34965432/>

<https://pubmed.ncbi.nlm.nih.gov/36113484/>

<https://pubmed.ncbi.nlm.nih.gov/37425903/> (selected example of emerging studies)

We thank the reviewer for this suggestion. We have now added examples of both mathematical and statistical modelling approaches to our section '*validating and advancing tissue maturity and functionality*' as additional methods that can be used to validate or inform on the criteria that advanced engineered breast models should meet (lines 561-572, page 24).

Referee #3 (Remarks for Author):

In the present review, the authors set the stage by discussing the biological significance of the mammary gland, its implications in breast cancer, and the limitations of current research models. They argue for the necessity of more complex and representative in vitro models to enhance understanding and treatment of breast health and pathology.

The authors summarized the current landscape of breast tissue engineering, including various approaches such as organoids, decellularized extracellular matrices, use of biomaterials, and biofabrication techniques (SLA, 3D bioprinting, etc) to mimic the native mammary gland environment. The authors also outlined requirements and challenges in creating predictive human breast models.

Next, the authors summarized potential applications of engineered breast tissues in preclinical studies, especially for improving understanding and treatment of breast cancer, and discussed the usage of human-derived materials in tissue engineering, and discussed the translational potential of these technologies into clinical practice, including therapeutic applications and personalized medicine.

In the discussion, the authors highlighted the need for more complex models that integrate intricate tissue architectures and cellular interactions, and discussed the potential of combining tissue engineering with other emerging technologies to enhance model fidelity and functionality.

Overall impression

The manuscript is well-structured, informative, and timely in the field of tissue engineering and mammary gland biology. It thoroughly discusses the current status, advancements, and challenges in creating representative models of human breast tissue, which are crucial

for understanding disease mechanisms and for the development of targeted therapies. The integration of multiple disciplines, including cellular biology, materials science, and bioengineering, is a notable strength of the review.

Strengths

- Comprehensive coverage of the biological and engineering aspects relevant to breast tissue engineering for both normal and malignant tissue.
- Inclusion of a wide range of references.
- Detailed review on the application of organ-on-chip and bioprinting technologies in breast tissue modeling.
- Specifically, Figure 3 is an efficient way for visualizing and comparing the different features and applications from multiple models.

We are very pleased that reviewer found our review informative and timely in the field of tissue engineering and mammary gland biology. We are very appreciative of the comments raised by the reviewer that have helped us to further improve our manuscript as detailed below.

Shortcomings are discussed below as comments

Line 69: Not only architecture is a limitation in organoid, but also "organoid-to-organoid" reproducibility in phenotype and cellular complexity, which hinders its application in drug screening. Reference: section "Towards a new generation of organoids". G. Rossi, A. Manfrin, M. Lutolf. Nature Reviews Genetics, 2018

We agree with the reviewer that also these factors hinder organoid experimentation and especially screening set-ups, which could be alleviated through tissue engineering approaches. This is also why we included, for instance, the work of Ao *et al.* (PNAS, 2022) that refers to the challenges of generating scalable, uniformly sized, and standardized organoids/spheroids for throughput assays and addresses some of these challenges through a pillar-lattice-array design, albeit for spheroids (lines 481-482, page 20). However, we agree that this is an important notion to also put forward where we introduce organoid technology, and we now added this and refer to the suggested review in the introduction of our revised manuscript (lines 83-85 on page 4).

Line 71: TEBs and TDLU are not the same. TEBs are transient structures found in the pubertal gland while the ducts elongate and may also be found in the pubertal breast in humans, see studies of the human breast by Barry

We thank the reviewer for pointing out this inconsistency. To avoid confusion and given the overall focus of our review on modelling approaches aimed at capturing the *mature* function of the human mammary gland, we have now removed the term TEB from this sentence. In

addition, we point out that TDLUs are considered the fully matured functional units of the human mammary gland (lines 95-96 on page 5). Later, where we talk more extensively about the ECM changes occurring during puberty, we now added that TEBs are transient structures that direct the growth of the duct (line 176-177 on page 8). In addition, we added a laymen definition of the TEB to our revised glossary (line 55, page 3).

Line 73f: myoepithelial cells should not be equated to basal cells. The former are defined based on the expression of contractile proteins the latter are defined by their position and comprise the former.

Again, we thank the reviewer for this careful notion and apologize for not being as sharp as we should have been in the use of this terminology. We have now revised this sentence (lines 98-99 on page 5) to clarify that the term myoepithelial cells only refers to basal cell that have adopted contractile properties.

line 80: better say "human breast" and "mouse mammary gland".

We now revised this in the entire manuscript, using breast when referring to human and mammary gland when referring to mouse.

Line 88: depending on the mouse strain with repeated estrous cycles some side branching or tertiary branching can be observed

We apologize for the confusion caused by our statement. We were not trying to highlight differences here between mouse and human in the extend of branching that can be observed. Rather, we wanted to point out that during development already a rudimentary tree structure is present that then further branches and evolves during postnatal development. To make this more clear, we have now removed "*while challenging to study in human, animal work has shown..*" from line 120 at page 6 of our revised manuscript.

Lines 84, 94, 110: lacking references

We apologize for this oversight and have now added the corresponding references.

The term "lobuloalveolar" is wrong. Lobe or lobules imply a separation within the tissue by sheets of connective tissue that do not exist in the mouse mammary gland. The human breast has lobes.

To avoid any confusion or ambiguity we now changed this term to alveologenesis (line 126, page 6 of the revised manuscript).

Lines 104 ff "This basement membrane, produced by myoepithelial cells and stromal fibroblasts, is a key regulator of the integrin-mediated polarized mammary epithelial phenotype."

This reviewer is under the impression that the epithelial cells also contribute to the basal lamina. Please, provide reference.

The basement membrane is produced by myoepithelial cells and stromal fibroblasts, and we now included references for both these cell types (line 165, page 7).

Line 146 floating collagen gels: the original work by Joan Emmerman should be quoted here.

We agree with the reviewer that this early work is important to recognize, which is also why we already included this paper in **Table 1** of our manuscript where we offer an overview of the effect of different ECM components and culture modalities on MEC differentiation and function. Per the reviewer's direction, we now also refer to this paper when we mention the use of floating gels in the main text of our manuscript (lines 220-221 on page 9).

Lines 175 ff However, because of interspecies differences, human tissue is needed to faithfully recapitulate the human mammary gland ECM, which is largely restricted by availability and ethical constraints."

This statement is misleading. Reduction mammoplasty tissue can be readily obtained under ethic protocols and indeed their use has lead to important lines of work starting with Margaret Stampfer.

We have nuanced this statement and in the revised manuscript now also refer to a published protocol from the Stampfer lab on how to process this material for readers interested in using it (lines 258-261, page 11/12). However, not all labs will readily have access to this type of clinical material and obtaining the required ethical protocols and approvals can be a lengthy process.

325 ff Section on drug discovery. Here interesting findings in different models are listed but it is not mentioned to what extent these were validated clinically. In other words is there any evidence that findings in these models can be extrapolated to in vivo settings?

The majority of data cited in this section was published in 2020, making the time frame too short for clinical validation. Bersini *et al.* used a microfluidic model to identify CXCR2-CXCL5 as a receptor-ligand pair involved in breast cancer bone metastasis, which was published in

2014. CXCR2 inhibitors are either still under clinical evaluation or have shown limited efficacy in clinical trials (Lazennec *et al.* Trends in Molecular Medicine, 2024). However, CXCR2 can interact with a multitude of ligands aside from CXCL5, so these trials do not specifically address the clinical outcome of blocking CXCR2-CXCL5 signaling. In support of the data obtained by Berisini *et al.*, similar findings have subsequently been published in an *ex vivo* bone culture system. We now refer to this work in the revised version of our manuscript (lines 431-432, page 18), to further substantiate the findings by Bersini *et al.*, supporting a role for CXCR2-CXCL5 signaling in breast cancer bone metastasis.

Introduce the engineering terms such as "extrusion-bioprinting", "pillar-lattice-arrays". Indeed, given the length of the review a schematic overview and comparison of the approaches that have been used to date will be helpful.

We addressed this comment in two ways. First, we added a glossary with the definition / explanation of these terms and explain them more extensively at the first mention in the text (e.g. lines 382-384 on page 16 and 394-395 on page 16). In addition, we added new **Figure 5** to the revised manuscript that contains a graphical representation of these main bioengineering technologies, lists their key features and advantages compared to the other technologies and includes the references to the papers that exploited these different methods for mammary gland modelling.

Line 356: In the section of "Understanding treatment resistance", the authors reviewed several works on the role of CAFs in rendering drug resistance to breast cancer in on-chip systems. However, this section would be broadened and more thorough if the authors can include works investigating other factors contributing to drug resistance on-chip, such as Cancer stem cells (CSCs), ECM remodelling, EMT, etc.

We agree with the reviewer that this section ended up quite dominated by findings on how CAFs impact breast cancer treatment outcomes, but that was not intentional. We performed a broad literature search on breast-on-chip systems being applied for drug response evaluations and the majority of these studies reported a critical role for CAFs in treatment resistance. The study by Langer *et al.* (Cell Reports, 2019) (lines 460-463, page 19), for instance, also included additional stromal cell types in their chip design; adipocytes and bone-marrow derived mesenchymal cells, and also analyzed collagen deposition. However, they only experimentally tested the impact of fibroblast secreted factors in treatment outcomes by supplementing with fibroblast conditioned medium. Work cited from Blanco-Fernandez *et al.* in this section did look at the effect of Collagen-I addition on chemotherapy response and we now included this finding in the revised version of our manuscript (page 467-468, page 19). In addition, we identified one additional paper that we now included that investigated ECM-mediated drug resistance in 3D cultures of breast cancer spheroids encapsulated in hydrogels (page 468-470, page 19). We hope that with these additions this

section now better acknowledges the role that the ECM can have in drug resistance, which we know also highlight in the concluding sentences of this section (line 475, page 19).

Engineered model dependent investigations into the progressive behavior of CSCs and their role in treatment resistance have up till now been largely restricted to microfluidics applied for single cell seeding (Lamichhane & Tavana, Cellular and Molecular Bioengineering, 2024). This is also the case for breast cancer, where this set-up was used to screen drugs that can reduce the outgrowth of single cancer cells into spheroids and, thus, inhibit their clone forming capacity (Lin *et al.*, Small, 2020). Tissue engineering models have not yet been applied to accommodate more complex 3D cultures for this purpose of studying the role of CSCs in treatment resistance. However, Dogan *et al.* in their review foresee a clear application of more structurally advanced tissue engineered models to study key features of CSC behavior and the interaction of these cells with drug candidates to understand their specific roles in treatment resistance (Dogan *et al.* Advanced Nanobiomed Research, 2022). We now include this perspective and refer to these papers in the revised version of our manuscript on page 18/19 (lines 445-454).

Line 391-393 sentence unclear

We now rephrased these sentences and also added more detail on the experimental details and read-outs used to offer a clearer description of the work performed in this paper (lines 491-497 on page 20).

Line 453: The authors could include the work from Alana Welm's laboratory, Nature Cancer, 2023, as another example of validation of BC model largely for triple negative breast cancers "While BC models can be validated by comparing tumor heterogeneity, gene signatures and invasive behavior to the original tumor biopsies, ultimate proof would come from comparing *in vitro* treatment responses to those observed in patients (Loewa *et al.*, 2023), and the use of PDOs, or other representative cell sources, will be key here."

And the work with intraductal xenografts Scabia *et al* Nat Com 2022.

We now refer to these papers in this section (lines 578-584, page 25) as example validations of breast cancer models, showing proof-of-concept that a drug response *in vitro* can be validated in the patient *in vivo* (Guillen *et al.* 2022) and highlighting the need for patient-specific models (Scabia *et al.* 2022). However, we also place this into context of further advanced and engineered *in vitro* breast cancer models that will also require validation of additionally modelled compartments, such as the TME or vasculature (lines 584-596, page 25).

Line 482 "endocrine hormonal" use one or the other

Corrected (line 623, page 26 of the revised manuscript).

Puberty and menstrual cycles should be shown as separate phases.

We have revised **Figure 2** of our manuscript to reflect puberty and menstrual cycles as separate critical stages involved in mammary gland development. In addition, in the accompanying manuscript text associated with this figure, we have provided more detail and references on both cell types and features involved in pubertal morphogenesis (lines 124-126 on page 6).

Line 499 By 2020, almost 30% of BC patients underwent either one-sided or double mastectomy to ensure removal of all tumor tissue, or prevent re-occurrence of BC (Fefferman et al., 2023).

Are you sure??? This sounds wild.

We checked these statistics again, and they are indeed correct: 70.7% of patients undergoing breast conserving surgery, whereas 18.1% underwent unilateral mastectomy and 11.3% of patients bilateral mastectomy in 2020. This is data from the national cancer database in the US and might, therefore, differ by region. However, while slightly older data, also in Europe reported mastectomy rates have been as high as 25% (Garcia-Etienne *et al.* European Journal of Cancer, 2012) and we now refer to this European statistic in the revised version of our manuscript as well (lines 642-643 on page 27).

The last section, as it is, does not add to the review. Materials that are introduced into the human body need to fulfill entirely different requirements than the different models and deserve a separate review.

We agree with the reviewer that material requirements will be very different for in-patient applications compared to *in vitro* modelling. However, our focus in this section is on the bioprinting and biofabrication technologies themselves that, while still in an early stage of development, are also being applied with the prospect of breast tissue reconstruction. Because of these efforts and the overall focus of our review on breast tissue engineering in health and disease, we think this is an active area of investigation and development worth highlighting to the reader, especially with mastectomy rates being as high as 30% and current reconstruction methods posing complications, including silicone leakage that especially received a lot of media attention in the last years. Therefore, this presents a specific implementation where breast tissue engineering could down the road reach profound clinical impact. This is why we think this is a section worth including and the heading of this subsection clearly informs the reader that this is an entirely different playing field compared to *in vitro* modeling efforts.

Problem with MDA MD 231 cells

We thank the reviewer for this careful notion. We have now adapted **Table 2** on page 23 of our revised manuscript accordingly to distinguish between studies using the MDA-MB-231-HTB-26 mammary gland-isolated adenocarcinoma line and those using pleural effusion-isolated metastatic breast cancer MDA-MB-231 cell lines.

Balance in Literature Presentation:

The literature review is thorough, but the authors could also show some conflicting findings or controversies within the field to provide a more balanced view.

More importantly, the authors should share more their points of view and including comparison, criticism, future perspectives, etc. Because "Writing a review article is about more than seizing on an interesting topic and gathering the relevant references, ... readers should learn something(s) that they couldn't get just by reading the references. Therefore, make sure you include your point(s) of view": <https://www.elsevier.com/connect/six-things-you-should-keep-in-mind-when-writing-a-review-article>

We agree with the reviewer and already included our opinion and vision on how the ultimate functional outcome of engineered breast tissue models should be analyzed and achieved. For instance, foreseeing a clear synergy with organoid technology (lines 538-540, page 22 and Conclusion & Discussion, page 28-30), depending on perfusable models to allow for temporally controlled growth factor and hormone supply and mechanical stimulation (lines 623-632, page 26/27 and Conclusion & Discussion), and requiring a broader analysis of essential milk components than is typically performed (lines 608-618, page 26). In addition, we stressed the importance of multi-material approaches and voiced criticism on the continued use of collagen-I dominated materials that not only neglects essential criteria offered by other critical ECM components, but also the fact that this induces invasive behavior by not providing a basement membrane required for functional polarization (lines 687-690, page 29).

To further adhere to the proposed review guidelines, we made additional changes in the revised manuscript, including:

- Offering perspective on how the use of engineered breast tissue could even go beyond modelling what is physiologically relevant, e.g. by creating an extra dense tumor stroma to select cellular therapies with advanced tumor penetrating properties (lines 506-510, page 20/21).
- Voicing opinion that interspecies differences in ECM composition need to be carefully considered when modelling human mammary gland function (lines 156-162, page 7).

- Providing perspective on the expected challenges encountered with breast cancer model validation (lines 599-603, page 25/26), especially relating to the introduction of a functional vasculature (lines 596-598 on page 25).
- Offering a broader comparison of read-out tools that can be applied for model validation, by including less commonly used mathematical and statistical modeling approaches (lines 561-572, page 24).

Summary tables and timelines:

Additional summary tables comparing the features of different model systems discussed could provide a quick reference for readers.

We now added new **Figure 5** to the revised manuscript, this figure provides a schematic overview of the main breast engineering model systems applied, lists their key features / benefits as compared to other engineering approaches, and includes the references to key papers developing or applying these technologies for breast tissue modelling.

Timelines summarizing the development of the main technologies or milestones in the field and their applications would enhance understanding and engagement.

Our goal was to create a timely overview of the current landscape of breast tissue engineering, giving a view on where the field is standing now and expected to be heading, without ignoring critical fundamental work performed in the past. Therefore, we feel that providing a historic timeline would be out of scope for this overall focus of the review and distracting from the most recent efforts in the field and expected upcoming advances. However, we did add new **Figure 5** presenting the key breast engineering technologies developed. Because it cites the main papers related to these technologies, this figure also gives insight into the timeframe of these developments.

Readability:

The review is generally straightforward and comprehensible albeit somewhat lengthy. The presence of long and fragmented sentences can disrupt the reading flow, resulting in an uneven reading experience.

For example, certain sentences could be clarified to enhance understanding: When they modeled two of the most common genetic alterations underlying the altered ErbB-PI3K-Akt signaling axis prominently occurring in BC; (new sentence started)

We now clarified the specific sentence mentioned by the reviewer, by splitting it and including more experimental details (line 414-418, page 17). In addition, we also carefully checked the entire manuscript for too long and, therefore, difficult to comprehend sentences and adjusted them (e.g. lines 64-69 on page 4 and lines 650-657 on page 27/28). Furthermore, to additionally help with the readability of our review and combat the lengthiness experienced by the reviewer, we have introduced additional sub-headings to break down larger text sections (e.g. line 120 on page 6, and on page 7 and 8).

The writing can be more efficient and concise while including more information for readers to comprehend and picture the literatures. For example, in line 390-391, description for hypoxia can be shortened by eliminating expressions of "microenvironmental stresses arising at the solid tumor core"; and in line 391-392 "In addition, their chip design could to some extent model NK cell exhaustion, which was not the case...", it would be more informative to also describe how the modeling of NK cell exhaustion is done, to what extent and why there seem to be a limitation. Similar lack of cleanness and elaboration are occasionally found in other parts of the review.

We thank the reviewer for this important feedback and now added more experimental details and the specifics on how exhaustion was measured in the section describing this paper (lines 491-497 on page 20). In addition, we checked other incidences where we could be clearer in our description of the literature and, for instance, changed this on lines 414-418 on page 17 of our revised manuscript.

12th Jun 2024

Dear Anne,

Thank you for submitting your revised manuscript. Almost everything is fine now, and I'll be able to accept your manuscript once the following minor editorial issues will be addressed:

- For more information: this section is meant to list relevant web links (as opposed to references). I would thus suggest to only keep in this section the following:

"Datasets of human breast (cancer) tissue and milk:

o Human breast cell atlases: <https://navinlabcode.github.io/HumanBreastCellAtlas.github.io/> and E-MTAB-13664, (<https://cellxgene.cziscience.com/collections/48259aa8-f168-4bf5-b797-af8e88da6637>) (Reed et al, 2024)

o scRNAseq of healthy tissue (GSE161892, GSE180878, GSE113197), BC tissue (GSE182694, GSE161529, GSE176078) and milk (http://bioinf.stemcells.cam.ac.uk:3838/khaled_wUFt1bHfmC/twigger/)

o Mass spectrometry data on milk (<https://glygen.org/global-search-result/Milk>) and HMOs (<https://glygen.org/publication/PubMed/20578730/>)"

- Tables: please move them after the figure legends.

- Figures: We work with Somersault1824 for graphics support for figures. They will assist with getting the figure to a publication ready state and will get in touch with you as soon as they have the first draft (~ 10 working days).

- Figure legends: could you please provide more details, in particular for Fig.1 and Fig.2?

- Please remove the blue font, and only keep in track changes mode any new modification.

Looking forward to receiving your revised manuscript,

With my best wishes,

Lise

The authors addressed the minor editorial issues.

14th Jun 2024

Dear Anne,

I am pleased to inform you that your manuscript is accepted for publication!
It will be sent to our publisher as soon as the figures are finalized and approved by you.

Your manuscript will be processed for publication by EMBO Press. It will be copy edited and you will receive page proofs prior to publication. Please note that you will be contacted by Springer Nature Author Services to complete licensing information.

There is no charge for this Review Article, but in a few weeks when you are contacted to sign your license agreement and review article proofs, please enter this token into the appropriate field in the Springer Nature author services system:
MTA3NJGXMZG3NG.

If you have any questions, please do not hesitate to contact the Editorial Office.
Thank you for your contribution to EMBO Molecular Medicine!

With my best wishes,

Lise
